# Paving the Way for Memory Enhancement: Development and Examination of a Neurofeedback System Targeting the Medial Temporal Lobe

**DOI:** 10.3390/biomedicines11082262

**Published:** 2023-08-13

**Authors:** Koji Koizumi, Naoto Kunii, Kazutaka Ueda, Keisuke Nagata, Shigeta Fujitani, Seijiro Shimada, Masayuki Nakao

**Affiliations:** 1Department of Mechanical Engineering, The University of Tokyo, Tokyo 113-8656, Japan; ueda@hnl.t.u-tokyo.ac.jp (K.U.); nakao@hnl.t.u-tokyo.ac.jp (M.N.); 2Department of Neurosurgery, The University of Tokyo, Tokyo 113-8655, Japan; nkunii@nsurg.jp (N.K.); kskngt@gmail.com (K.N.); sfujitani@gmail.com (S.F.); shimajisan+work@gmail.com (S.S.)

**Keywords:** neurofeedback, memory enhancement, medial temporal lobe, intracranial electrode, bidirectional control, memory encoding, intracranial electroencephalogram, epilepsy

## Abstract

Neurofeedback (NF) shows promise in enhancing memory, but its application to the medial temporal lobe (MTL) still needs to be studied. Therefore, we aimed to develop an NF system for the memory function of the MTL and examine neural activity changes and memory task score changes through NF training. We created a memory NF system using intracranial electrodes to acquire and visualise the neural activity of the MTL during memory encoding. Twenty trials of a tug-of-war game per session were employed for NF and designed to control neural activity bidirectionally (Up/Down condition). NF training was conducted with three patients with drug-resistant epilepsy, and we observed an increasing difference in NF signal between conditions (Up–Down) as NF training progressed. Similarities and negative correlation tendencies between the transition of neural activity and the transition of memory function were also observed. Our findings demonstrate NF’s potential to modulate MTL activity and memory encoding. Future research needs further improvements to the NF system to validate its effects on memory functions. Nonetheless, this study represents a crucial step in understanding NF’s application to memory and provides valuable insights into developing more efficient memory enhancement strategies.

## 1. Introduction

Several studies targeting patients with lesions in the medial temporal lobe (MTL) have reported impairments in short-term memory function [1,2,3,4,5]. Recent research in rats indicates that the coordinated neural activity patterns between the dentate gyrus and the hippocampal CA3 region contribute to working memory [6]. Additionally, it was reported that damage to the right MTL may lead to a heightened propensity for impairments in spatial memory, integral to navigation and self-location, while damage to the left MTL may increase deficits in verbal memory, essential for memorising auditory and visual linguistic information [7].

In recent years, electrical stimulation to deep brain regions has been established, leading to an increase in studies aiming to enhance memory targeting the MTL [8]. However, some studies specifically targeting the hippocampus have reported a decline in memory [9,10,11,12]. This decline is hypothesised to result from an acute depolarization block leading to memory impairment [9]. Akin to electrical stimulation, neurofeedback (NF) is one of the techniques for modulating brain function. NF converts brain activity into perceptible information, such as bar length or circle size, and provides feedback, allowing individuals to self-regulate their brain activity. Through repeated NF training, it is expected that the regulation is facilitated of cognitive functions associated with brain activity. NF has gained momentum in clinical applications, with reports of symptom alleviation in various mental disorders, such as anxiety, depression, schizophrenia [13], and ADHD [14]. Additionally, there has been an increase in opportunities for NF training in healthy individuals to enhance memory [15], meditation [16], and attentional focus [17]. Since NF relies on the brain’s autonomic learning capability to regulate neural activity, it offers a potentially lower risk of interfering with memory function compared to electrical stimulation. Moreover, anatomical evidence reveals that memory function involves not only the hippocampus but also multiple regions, such as the thalamus, mammillary bodies, and the cingulate gyrus [18]. In order to regulate such complex neural networks, we believe that a closed-loop control system that relies on the brain’s autonomic learning capability, such as NF, would be more rational and effective compared to open-loop control systems that simply deliver external electrical stimulation.

Patients with drug-resistant epilepsy, undergoing intracranial electrode placement through craniotomy for the purpose of epileptic focus diagnosis, provide an opportunity for intracranial electroencephalogram (iEEG) measurements. Recently, there has been increased interest in applying this invasive brain activity measurement method to brain-computer interfaces [19]. Signals obtained from intracranial electrodes offer superior temporal resolution compared to functional magnetic resonance imaging (fMRI) and reduce the impact of artefacts compared to scalp EEG. Additionally, the signals are not attenuated by the dura mater, skull, or scalp tissue. These advantages can be beneficial in NF studies, such as those focusing on modulating somatosensory–motor function [20,21,22] or emotional function [23]. However, these approaches are still in their early stages, and there are few NF studies specifically targeting memory function in the MTL [24].

### 1.1. Learning Mechanisms in Neurofeedback

The mechanism behind NF training is believed to be rooted in operant conditioning [25]. Operant conditioning involves changing the occurrence probability of a response under a specific condition by providing a consequence (reinforcement or punishment) in response to voluntary behaviour (operant behaviour). The “specific condition” serves as an antecedent that functions as a cue for the voluntary response. Operant conditioning involves learning the association, known as the three-term contingency, between antecedent, behaviour, and consequence.

In the context of NF, let us consider an illustrative scenario where the user can increase the length of a bar displayed on the screen when a specific brain activity pattern occurs during a specific cognitive condition. Through NF training, as the user engages in trial and error, they may accidentally produce the desired brain activity pattern and achieve success in increasing the length of the bar. Through repeated these experiences, the user’s probability of generating the desired brain activity pattern in a specific cognitive condition gradually increases. In other words, in NF, the user learns the contingency between the specific cognitive condition (antecedent), the brain activity pattern (behaviour), and the success in bar length adjustment (reinforcement).

The presence of consequence is crucial for the success of operant conditioning. If the reinforcements or punishments following operant behaviour are insufficient, the changes in the occurrence probability of the response will be weak, making it difficult for learning to take place. A neurophysiological theory of NF suggests the significant contribution of the striatum, a part of the reward system, to NF [26], and indeed, several NF studies have reported the observed involvement of the striatum [27,28,29]. These emphasise the importance of the consequence in NF too.

### 1.2. Challenges in Neurofeedback Research on Memory Function

#### 1.2.1. Challenges in Brain Activity Measurement

NF studies targeting memory have extensively used scalp EEG [15,30,31,32,33,34,35,36,37]. Memory function has long been suggested to be associated with theta oscillations [38,39,40], and an increase in theta power during memory encoding has been reported in many EEG studies [38,41,42]. Based on the accumulation of such foundational research, many memory EEGNF studies have been implemented for increasing theta power. So far, previous studies have shown that upregulation of theta activity in the frontal midline improves control processes during memory retrieval [31] and accelerates and enhances memory consolidation [33,36]. Studies utilising intracranial electrodes have also reported increased theta power in the MTL during successful encoding and retrieval [43,44,45,46,47,48,49], and increased theta phase synchronization between the hippocampus and other cortical regions (rhinal cortex, prefrontal cortex) during memory encoding [50,51]. However, some studies also report a decrease in MTL theta power during encoding [52,53,54,55,56,57,58], and the causal relationship between MTL theta power changes (increases or decreases) and memory function (encoding success or failure) is inconsistent across studies. In our previous study [24], we conducted NF using iEEG of MTL and reported an increase in theta power in one participant; however, no accompanying changes in memory function were observed.

#### 1.2.2. Challenges in Modulating Memory Function

In NF studies aiming to improve mental disorders, it is common to conduct NF during a resting state without imposing additional tasks on participants. This is because the resting state already corresponds to the “specific cognitive condition” in the learning mechanism of NF. In other words, in mental disorders, the chronic symptoms typically persist even during the resting state, which is why NF in this state enables the learning of three-term contingency. The same applies to healthy individuals training their attention, relaxation, or meditation using NF. On the other hand, when conducting NF training for memory function, the “specific cognitive condition” corresponds to the condition requiring memory function. Therefore, it is essential to create a situation that necessitates memory function, such as imposing memory tasks on participants, and conduct NF within that context to efficiently learn the three-term contingency and achieve successful memory NF. Traditional memory NF studies have often required several weeks or more for participants to gain self-regulation of brain activity [15,59]. We believe this may be attributed to the fact that they were conducted during resting states, without creating a situation that necessitates memory function. Additionally, during the resting state, participants lack relevant clues on how to control their brain activity to adjust the feedback signal, leading to higher difficulty in learning self-regulation and requiring an extended time for the effects to appear. If NF can be conducted within a context that necessitates the desired brain function, participants will have clues on how to change the feedback signal, potentially making it easier to acquire self-regulation of brain function in a shorter period.

However, when participants perform tasks requiring a specific brain function while simultaneously receiving feedback, dual-task interference occurs, diverting attention away from the feedback, making self-regulation difficult, and vice versa. To address this issue, a task-based NF approach has been proposed [60,61], where the periods for task performance and feedback are separated, allowing for alternating short periods of task engagement and feedback. This approach enables participants to engage in tasks and feedback without divided attention. In this study, we adopt this task-based NF approach to investigate whether the neural activity in the MTL associated with memory function can be modulated.

### 1.3. Aims

This study aims to develop a task-based NF system, incorporating memory tasks and NF, and to observe the control of neural activity and resulting effects on memory function through NF training conducted to modulate MTL neural activity in both upward and downward directions.

## 2. Materials and Methods

### 2.1. Participants

In this study, we targeted three patients with drug-resistant epilepsy (P01, P02, P03) who underwent intracranial electrode implantation for clinical purposes at the University of Tokyo Hospital. The characteristics of each participant are shown in Table 1. The Full-Scale Intelligence Quotient (FIQ) was evaluated using the Wechsler Adult Intelligence Scale-fourth edition (WAIS-IV). Language dominance was assessed using functional MRI or cortical electrical stimulation via subdural electrodes during a language task [62], confirming that the language-dominant hemisphere for all participants was the left hemisphere. Since language dominance and language memory dominance are often concordant [63], it was assumed that language memory dominance is also located in the left hemisphere. Prior to the surgery, all participants underwent the Japanese version of the Wechsler Memory Scale-Revised (WMS-R), with a mean of 100 and a standard deviation of 15. The index score for P01 was 51, confirming that verbal memory function was impaired, which is speculated to be due to sclerosis in the left hippocampus. The verbal memory of P02 and P03 was found to be within the normal range.

This study was conducted with the approval of the Institutional Review Board of the University of Tokyo Hospital (Approval No. 1797) and in accordance with the Declaration of Helsinki. All participants received sufficient explanations regarding the purpose and content of the study and provided written informed consent.

### 2.2. Acquisition of iEEG Data

The placement of intracranial electrodes was determined based on clinical purposes, which involved identifying the epileptogenic zone for each participant. The respective electrodes used and their placement locations for each participant are described in the results section. iEEG data were recorded using gUSBamp (gTec, Schiedlberg, Austria) at a sampling frequency of 512 Hz. When acquiring iEEG data, a bandpass filter of 0.1–200 Hz and a notch filter at 50 Hz were applied using the settings of g.HIsys from g.tec Suite 2020 to reduce power line noise. The signal from the reference electrode (see Section 3.1.1, Section 3.2.1, and Section 3.3.1) was used as a reference during signal acquisition and online analysis.

### 2.3. Data Acquisition Procedure

Prior to participating in the study, the participants watched an explanatory video and read an instruction manual to ensure a sufficient understanding of the tasks. The participants were seated in an electrically shielded room and instructed to look at a monitor screen displaying visual stimuli during the tasks. The monitor screen was divided into upper and lower sections by a central line. The upper screen displayed instructional text and the memory task, while the lower screen displayed NF (Figure 1a).

At the beginning of each session, the participants were instructed to be relaxed while fixating on a fixation point displayed in the centre of the upper screen for 20 s. The subsequent memory NF paradigm consisted of 20 trials per session, with each trial comprising a memory task (encoding period followed by recognition period) and an NF period (Figure 1b). During the encoding period, five words were sequentially presented in the centre of the upper screen for 1.6 s each, with a 2.6-s interval between them. The participants were required to memorise these words. In the subsequent recognition period, a single word was presented on the upper screen, and the participants had to determine whether the word was one of the five words presented during the encoding period (task 1) and, if so, what number they saw (task 2). The question screens for those tasks were presented sequentially for 8 s each, and the participants provided their responses using key inputs. In the following NF period, the neural activity in the medial temporal lobe during the encoding period was visually presented as feedback on the lower screen in the form of a tug-of-war game. The duration of each session was set to approximately 15 min, considering the physical and mental fatigue of the patients.

### 2.4. Memory Task

The words selected for the memory task were obtained from the Word List by Semantic Principles (WLDP) version 1.0, provided by the Centre for Language Resource Development, National Institute for Japanese Language and Linguistics [64]. This lexical database comprises 96,557 words and employs crowdsourcing and Bayesian linear mixed models to estimate word familiarity [65]. Initially, Japanese hiragana or katakana words consisting of three letters or three syllables were extracted to form multiple sets of six words. The word combinations were adjusted to ensure an equal average familiarity score across the sets. Each session included 20 word sets, with one set assigned per trial. The 20 word sets were extracted so that they would not overlap across sessions. 

During the encoding period, five out of the six words in each set were presented. In the subsequent recognition period, one of the five words presented during encoding was displayed as an “old” stimulus, while the remaining word not presented during encoding was shown as a “new” stimulus. Out of the 20 trials, 10 trials featured the presentation of “old” stimuli and the other 10 trials featured the presentation of “new” stimuli. The sequence of trials, with either “old” or “new” stimuli, was randomised.

Regarding task 1, the accuracy and recall scores were calculated. The accuracy score is the probability of a participant correctly discerning whether a presented stimulus is “old” (previously encountered in the encoding period) or “new” (not previously seen during the encoding period). This measure provides an overall view of the participant’s ability to accurately identify both old and new stimuli. Conversely, the recall score solely focuses on the “old” stimuli. This is the probability of correctly identifying a stimulus as “old” when it was indeed presented earlier during the encoding period. Contrary to the accuracy, the recall does not consider the identification of “new” stimuli. As such, it exclusively assesses the participant’s ability to recall or recognise previously presented stimuli. For task 2, the accuracy score (i.e., the probability of correctly identifying the order in which the “old” stimulus was presented in task 1) was computed. As an example, in the trial depicted in Figure 1b, “Tuna” was the first word presented among the five, indicating that the correct answer for task 2 is “first”.

### 2.5. Neurofeedback

Real-time analysis and feedback of the acquired iEEG data were performed using a custom program developed with g.HIsys (gTec, Schiedlberg, Austria) and MATLAB R2020a (MathWorks, Natick, MA, USA). During the memory encoding period of 13 s ((1.0 + 1.6) s × 5 words), iEEG data were extracted, and an artifact subspace reconstruction [66,67] (ASR) was applied to reduce epileptic spike-related artifacts. The ASR parameters were adjusted based on the extent of epileptic spike contamination. Specifically, the proportion of brain wave potentials exceeding the mean ± 3 standard deviations for 13 s was defined as the occurrence rate of interictal activity (RIA), and waveform correction was performed with k = 10 for <1%, k = 6 for between 1% and 1.5%, and k = 4 for >1.5%. Subsequently, the 12 s of iEEG data, excluding the initial 1-s period in which the encoding words were not presented, were used to calculate the ratio of low theta band power (2–6 Hz) to the total power (0.5–32 Hz). The average ratio of low theta band power (LTR) across four electrodes was then computed. For each trial (i), the relative magnitude of LTR_i_ to LTR_1_ (LTR_i/1_ = 100 × LTR_i_/LTR_1_) was used as the neurofeedback (NF) signal. The range of 2–6 Hz encompasses the high delta and the low theta by this standard classification (delta, 0.5–4 Hz; theta, 4–8 Hz). However, in the field of human memory research, the frequency band related to memory processing and similar functional characteristics to theta oscillations (4–8 Hz) observed in rodents which have been noted, is often shifted to a lower frequency range and referred to as ‘low theta’ or ‘slow theta’ [45,47,68]. For instance, Goyal et al. suggested that theta oscillations in the hippocampus can be divided into high- and low-frequency theta oscillations, which may, respectively, reflect spatial and non-spatial cognitive processes [69]. Thus, in our study focusing on verbal memory, following these previous studies, we have elected to classify the 2–6 Hz range as the low theta band. In addition, in this study, we employed a frequency band ranging from 0.5 to 32 Hz for the computation of total power, excluding any frequency bands above 32 Hz. According to previous studies, high gamma bands over 32 Hz have been reported to be implicated in memory functions, similar to theta [24,57,70]. However, here, as we particularly focused on the low theta band as a frequency band related to memory encoding, we determined that including frequencies above 32 Hz in the total power computation could potentially influence neurofeedback.

We employed a bidirectional control NF system and introduced gamification and reward elements. The reason for adding gamification and reward elements is that attention, motivation and mood affect efficient learning and success in NF [71]. Specifically, we employed a tug-of-war game with 20 trials per session, where a larger LTR_i/1_ value caused the rope to be pulled towards the right (red team side), while a smaller LTR_i/1_ value resulted in the rope being pulled towards the left (white team side), with the marker moving right and left accordingly (Figure 2a). Participants were instructed that their brain activity during word memorization would be utilised for a tug-of-war game, and the outcome would be determined after each trial. The participants were also informed about which team to root for before the start of each session (Up/Down conditions based on the direction of adjusting the LTR_i/1_ value) and were encouraged to strive to win trial and error. The Up and Down conditions were alternated between sessions. For P01 and P03, odd-numbered sessions were assigned to Up and even-numbered sessions to Down, while for P02, it was the reverse. The research conductors did not provide explicit instructions regarding mental strategies to the participants, and the correspondence between the Up/Down conditions and the winning/losing teams was blinded. Additionally, the research conductors and data analysts were separate individuals, and the analysts were blinded to the information regarding which condition was used in each session. When the supported team won, 10 points were added to their score. Moreover, if the LTR_i_ changed by more than 20% from LTR_1_ and the supported team won, the score was doubled (20 points), accompanied by the playback of cheering audio. Conversely, if the supported team lost, 10 points were added to the opponent’s score, and if the LTR_i_ changed by more than 20% from LTR_1_ and resulted in the loss of the supported team, the opponent’s score was doubled (20 points). A score table was displayed at the top of the tug-of-war screen, allowing the participants to monitor the cumulative scores for their supported team as the trials progressed. Figure 2b illustrates the transition of the NF screen from the encoding period to the NF period in trial i of sessions rooting for the red team (Up condition). After completing all 20 trials, the outcome of each session was determined based on the cumulative scores. When the supported team won, a celebration screen and a loud cheer were displayed as audio. In the case of a draw, only a notification was displayed informing the result of a draw. In case of loss, a screen indicating disappointment was shown.

### 2.6. Offline Data Analysis

To examine how the difference in NF signal values (LTR_i/1_) between the Up and Down conditions changed before and after NF training, the difference in median NF signal values between trials was calculated for each condition (Up and Down) in both the first and final sessions. Subsequently, these differences were compared between the first and final sessions. 

Next, the transition of NF signal values (LTR_i/1_) between sessions and within sessions was examined for each participant. The transition between sessions for LTR_i_ was also investigated. Statistical analyses were conducted to determine if there were significant increases or decreases in the transition of these values between sessions. Nonparametric tests were employed for all statistical analyses conducted across all participants due to the lack of normality in the distribution of some data, as assessed using the Shapiro-Wilk test. The significance level was set at α = 0.05. For each session, the median of LTR_i/1_, which represented the NF signal values from trial 2 to trial 20, was compared to the baseline, the LTR_1/1_ value (100) of trial 1, using the two-sided Wilcoxon signed-rank test. Additionally, the Wilcoxon rank-sum test was employed to examine if there were differences between the conditions at the same session number, and if there were changes across sessions within the same condition. The effect size (*r*) was calculated as the Z-statistic divided by the square root of the sample size (N). Similarly, the Wilcoxon rank-sum test was used to examine whether there were differences in RIA between sessions. Spearman’s rank correlation was utilised to investigate the associations between the changes in the three scores obtained from the recognition tasks and the changes in physiological data (NF signal values, LTR, and RIA). For the correlation analysis, only the trials relevant to each score were selected, and their median NF signal values and median LTR values were calculated for each session. Multiple comparisons were adjusted using the Holm–Bonferroni method, with a significance level set at α = 0.05. Furthermore, since EEG tends to exhibit larger power in lower frequencies, there is a possibility that the delta band (0.5–2 Hz), which is even lower in frequency than the targeted low theta band for NF, may influence the transitions of LTR. Therefore, we investigated the transitions of the total power (0.5–32 Hz), low theta power, and delta power between sessions.

To examine the frequency and temporal characteristics of neural activity during memory encoding, a time–frequency analysis was performed. The iEEG power spectrum (obtained through fast Fourier transform) and power spectrum using complex Morlet wavelets ($e^{i2\pi tf}e^{-t^{2}/(2*\sigma^{2})}$, *t*: time, *f*: frequencies increasing logarithmically from 1 Hz to 64 Hz with 31 log-spaced steps, *σ = n/2πf*: bandwidth of each frequency band, *n*: wavelet cycle numbers increasing logarithmically from 4 to 10) were multiplied for each trial data of each electrode. Then, the inverse fast Fourier transform (convolution in the frequency domain) was applied to obtain the time–frequency representation. The squared magnitude of the convolution results was defined as the power estimate at each time point and frequency band. After that, the power estimates from the four electrodes were averaged, resulting in time–frequency features for each trial. To investigate the time–frequency characteristics related to memory encoding, trials in each session were divided into correct trials (Correct) and incorrect trials (Error) based on the accuracy of task 2. Specifically, the median of each Correct/Error trial was calculated for each session. Then, the average power estimates from −600 ms to −100 ms before the presentation of the first word were considered as the baseline activity. Using this baseline activity, the power estimates during memory encoding were Z-transformed and normalised for each condition. By calculating the average of the Z-scores across sessions for Correct and Error trials, the time-frequency characteristics of each condition were obtained. Furthermore, the difference in time–frequency characteristics between Correct and Error trials was assessed by subtracting the Z-scores of Error trials from Correct trials for each session and calculating the average across sessions. It is worth noting that statistical comparisons were not conducted due to constraints on the number of sessions.

Similarly, the time–frequency characteristics of the two conditions (Up/Down) were compared during the two NF training stages (first/final). For each combination of condition and stage, the median values of trials, excluding trial 1 and trials contaminated with noise, were calculated. Then the average power estimates from −600 ms to −100 ms before the first word presentation were used as the baseline activity and underwent Z-score transformation. By standardisation, the time-frequency characteristics of each condition were obtained. The differences between the two groups (condition/stage) were statistically compared using nonparametric randomization tests. In the randomization tests, first, the time-frequency maps of each trial were shuffled between the two groups, and the test statistic *t* was obtained for each time–frequency pixel. By performing 1000 shuffles, the mean and standard deviation of the test statistics *t* for each pixel were calculated. The actual *t*-values were then converted to z-scores based on the mean and standard deviation from the shuffling procedure, creating a time–frequency map representing the differences between the two groups. Cluster-based multiple comparison corrections were applied to identify statistically significant spatiotemporal clusters. In this process, the maximum cluster size formed by pixels exceeding the threshold (α = 0.05) of the *t*-distribution in each shuffle out of 1000 shuffles was extracted to obtain the null distribution of cluster sizes. Subsequently, the threshold was set at the 95th percentile of this null distribution, and clusters of pixels exceeding this threshold were considered statistically significant as large clusters.

## 3. Results

There were no participants who experienced excessive sleep deprivation, nor did sleep deprivation lead to seizure induction in any participant. All participants remained conscious, and none reported feeling unwell during the training.

System errors or significant noise contamination led to the exclusion of trials or sessions where proper NF could not be achieved. Additionally, trials in which key input was missed, resulting in an inability to confirm responses for the recognition task, and trials where effective NF was not possible were excluded during the calculation of recognition performance.

When comparing RIA among participants using the Kruskal–Wallis H test, differences were observed among P01, P02, and P03 (*H* = 79.103, *p* < 0.001). Subsequent post-hoc analysis with the Steel–Dwass test indicated that P03 had significantly higher rates compared to both P01 (*W* = 10.318, *p* < 0.001) and P02 (*W* = 10.800, *p* < 0.001). However, no significant difference was observed between P01 and P02 (*W* = 0.240, *p* = 0.984).

The difference in NF signal values for the Up condition relative to the Down condition (Up–Down) is shown in Figure 3. In the first session, the NF signal difference varied among participants, ranging from positive values for P01 (27.35%) to negative values for P03 (−52.66%), with P02 having a value of 1.90%. In the final session, all participants exhibited positive NF signal differences and showed larger values compared to the first session, with P01 at 42.35%, P02 at 40.25%, and P03 at 6.82%.

The following section provides the results for each participant.

### 3.1. Results for P01

P01 completed four sessions of memory NF training over a span of five days. P01 did not experience any seizures during or within the 24 h before and after the training. For RIA, no significant differences were found between sessions after adjusting for multiple comparisons. After the NF training, P01 underwent surgery to remove the left temporal lobe, resulting in the cessation of seizures. Post-surgery, the verbal memory Index Score on the WMS-R was below 50 and showed a decline compared to pre-surgery levels.

After the end of session 2, it was reported that, in the course of trial and error in the NF, P01 adopted a strategy of randomly answering the recognition task. Thus, we asked P01 to avoid random responses from sessions 3 onwards. There were no mentions of mental strategies from session 3 onwards. Furthermore, in session 3, due to a system error, the baseline for the NF signal was referenced not from the encoding period data of trial 1, but from a period preceding it with higher LTR values (specifically, 153.81% when LTR1 was set as 100%). Consequently, in session 3, NF was conducted under conditions that made it difficult for the red team to win. Since NF was consistent with LTR values in the MTL, it was not excluded from the analysis. However, careful interpretation of the results in session 2 and 3 is required.

#### 3.1.1. Intracranial Electrodes Used for Memory NF (P01)

In P01, deep electrodes (Unique Medical, Tokyo, Japan) were implanted in the right MTL. The iEEG data were measured from four platinum electrodes (1 mm in length), positioned at 5 mm intervals (centre to centre) from the tip located in the right hippocampus, and used for NF (Figure 4). A reference electrode was placed subcutaneously on the right side for P01.

#### 3.1.2. Performance Changes in the Recognition Task (P01)

The session-to-session changes in the three scores for the recognition task are shown in Figure 5. The accuracy of task 1 significantly decreased in Session 2, where random responding occurred. The recall score was equivalent to Sessions 3 and 4. Comparing Up and Down conditions, both in the first sessions (Session 1 vs. 2) and final sessions (Session 3 vs. 4), the accuracy of task 2 was higher for Up than for Down, and the difference increased in the final sessions.

#### 3.1.3. NF Signal Changes across Sessions (P01)

The session-to-session changes in the NF signal values (LTR_i/1_) are presented in Figure 6. The NF signal values were significantly higher for Up in Session 1 (*Mdn* = 124.81, *n* = 19, *z* = 4.118, *p* < 0.001, *r* = 0.945) and Session 3 (*Mdn* = 137.81, *n* = 18, *z* = 4.475, *p* < 0.001, *r* = 1.055) compared to the baseline (LTR_1/1_), with large effect sizes. However, there were no significant changes in the Down condition in Session 2 (*Mdn* = 95.82, *n* = 18, *z* = −1.835, *p* = 0.067, *r* = 0.432) and Session 4 (*Mdn* = 95.45, *n* = 16, *z* = −1.197, *p* = 0.231, *r* = 0.299) compared to the reference. When comparing Up and Down conditions for the same session number, both the first session (*z* = 4.467, *p* < 0.001, *r* = 0.734) and second session (*z* = 4.934, *p* < 0.001, *r* = 0.846) showed significantly higher values for Up than for Down, with large effect sizes. In the Up condition, the NF signal values significantly increased with each session, showing a large effect size (*z* = 3.312, *p* < 0.001, *r* = 0.545). However, there were no significant changes between sessions in the Down condition (*z* = 0.035, *p* = 0.986, *r* = 0.06).

The similarity between the performance changes in task 2 and the NF signal changes was observed. However, the results of the statistical analysis showed no significant correlations between the performance on the recognition task and the NF signal values for task 1 accuracy (*r* = 0.80, *p* = 0.33), recall (*r* = 0.63, *p* = 0.37), and task 2 accuracy (*r* = 1.00, *p* = 0.08). Furthermore, no significant correlations were found with other physiological data for task 1 accuracy (LTR, *r* = 0.00, *p* = 1.00; RIA, *r* = 0.32, *p* = 0.68), recall (LTR, *r* = 0.32, *p* = 0.68; RIA, *r* = 0.63, *p* = 0.37), and task 2 accuracy (LTR, *r* = 0.80, *p* = 0.33; RIA, *r* = 0.20, *p* = 0.92).

#### 3.1.4. NF Signal Transition within Sessions (P01)

Regarding the first and final sessions, we compared the within-session transitions of NF signal values between Up and Down conditions (Figure 7). Both in the first and final sessions, the NF signal values exhibited higher overall trends in the Up condition compared to the Down condition across trials. Additionally, in the Up condition, there was a steeper rise in NF signal values from trial 1 during the final session compared to the first session.

#### 3.1.5. Low Theta Power Ratio (LTR) Changes across Sessions (P01)

The LTR_1_, which served as the baseline, showed approximately a 10% difference across conditions: 33.7% in Session 1, 43.4% in Session 2, 32.8% in Session 3, and 41.6% in Session 4. A comparison between conditions was performed for the LTR_i_ values from Trial 2 onwards, similar to the NF signal analysis (Figure 8). When comparing Up and Down conditions for the same session number, there were no significant differences in the first session (*z* = −0.091, *p* = 0.940, *r* = 0.015). However, in the second session, Up showed significantly higher values than Down, with a moderate effect size (*z* = 2.691, *p* = 0.006, *r* = 0.462). In the Up condition, the LTR_i_ values significantly increased with each session, showing a moderate effect size (*z* = −2.553, *p* = 0.01, *r* = 0.419). However, there were no significant changes between sessions in the Down condition (*z* = 0.897, *p* = 0.384, *r* = 0.154).

#### 3.1.6. Frequency Band Power Changes across Sessions (P01)

The transitions between sessions of total power, delta power, and low theta power are presented in Figure 9. Both delta power and low theta power had similar median magnitudes. All frequency band powers showed a trend of higher median values in the Up condition compared to the Down condition. The patterns of total power and low theta power changes exhibited similarities to the NF signal changes and the performance changes in task 2. 

#### 3.1.7. Time-Frequency Maps for Correct and Error Trials (P01)

The time–frequency maps for Correct and Error trials are shown in Figure 10a,b. Both maps exhibited intermittent power increases in the theta and alpha bands. When comparing Correct to Error trials, it was observed that during the mid-phase of the encoding period, the low theta band power increased before stimulus presentation and decreased during stimulus presentation (Figure 10c). 

#### 3.1.8. Time-Frequency Maps for Up/Down and First/Final Sessions (P01)

The time-frequency maps for the Up and Down conditions, as well as the first and final sessions, along with the difference maps between sessions, are presented in Figure 11. In comparison between the Up and Down conditions, in both the first and final sessions sustained power increases in the gamma band were observed for the Up condition compared to the Down condition. Additionally, intermittent power increases in the alpha and beta bands during the middle of the encoding period were observed for the Down condition in the first sessions. However, in the final sessions, the Up condition exhibited more consistent power increases from the low theta to beta bands. When comparing the first and final sessions, power increases in the Up condition and power decreases in the Down condition from the low theta to beta bands were observed during the middle of the encoding period. Furthermore, in the Up condition, a significantly large cluster indicating a decrease in delta band power at the end of the encoding period was observed for the final session compared to the first session. 

### 3.2. Results for P02

P02 completed six sessions of memory NF training over a span of five days. There were brief partial seizures on the evening before Session 3 and a few hours before Session 4, 5, and 6. However, there were no residual symptoms during the training. For RIA, both Session 1 (*z* = 3.389, *p* < 0.001, *r* = 0.557) and Session 2 (*z* = 3.231, *p* < 0.001, *r* = 0.524) showed significantly higher rates compared to Session 6, even after Holm–Bonferroni correction. After the series of NF training, P02 did not undergo surgery; therefore, the WMS-R assessment was not conducted post-surgery. 

During sessions 1 and 2, it was reported that P02 was occupied with memorising the words. After sessions 5 and 6, reports indicated that the white team tended to win when P02 memorised the words with relaxation without concentrating too much. In contrast, when P02 concentrated hard in order to memorise, the red team seemed to win. Regarding session 4, it was observed that artificial noise with high amplitude was mixed during trial 1, making it difficult to adjust noise reduction using ASR. Since the NF signal values used trial 1’s LTR as the baseline, session 4 was excluded from the analysis, and any trials with confirmed artificial noise contamination were also excluded.

#### 3.2.1. Intracranial Electrodes Used for Memory NF (P02)

In P02, subdural electrodes (Unique Medical, Tokyo, Japan) were implanted in the left MTL, covering the hippocampus. Four platinum electrodes (1.5 mm in diameter) were longitudinally placed at 5 mm intervals (centre to centre) along the left parahippocampal gyrus, and iEEG was measured and used for NF (Figure 12). A reference electrode placed on the dural side of the right temporal lobe was used for P02.

#### 3.2.2. Performance Changes in the Recognition Task (P02)

The session-to-session changes in the three scores for the recognition task are shown in Figure 13. The recall score for task 1 slightly decreased from session 1 to 2, but all aspects of the task showed improvement from session 1 to 3 and a decline from session 5 to 6.

#### 3.2.3. NF Signal Changes across Sessions (P02)

The session-to-session changes in NF signal values (LTR_i/1_) are presented in Figure 14. The NF signal values were significantly higher for Up in session 2 (*Mdn* = 153.49, *n* = 19, *z* = 4.621, *p* < 0.001, *r* = 1.060) compared to the baseline (LTR_1/1_), with a large effect size. However, in session 6, the NF signal values for Up were not significantly higher (*Mdn* = 117.91, *n* = 17, *z* = 2.216, *p* = 0.027, *r* = 0.538). For Down, the NF signal values were significantly higher in session 1 (*Mdn* = 151.58, *n* = 18, *z* = 3.800, *p* < 0.001, *r* = 0.896) compared to the baseline, with a large effect size. In session 3, there were no significant changes (*Mdn* = 101.37, *n* = 19, *z* = 0.00, *p* = 1.00, *r* = 0.00), while in session 5, the NF signal values were significantly lower compared to the baseline, with a large effect size (*Mdn* = 77.65, *n* = 19, *z* = −3.834, *p* < 0.001, *r* = 0.880). When comparing Up and Down conditions for the same session number, there were no significant differences in the first session (*z* = 0.091, *p* = 0.940, *r* = 0.015). However, in the final session, Up showed significantly higher values than Down, with a large effect size (*z* = 4.009, *p* < 0.001, *r* = 0.668). It is noteworthy that, in the Up condition, there was a significant decrease in NF signal values as sessions progressed, with a large effect size (*z* = 3.565, *p* < 0.001, *r* = 0.594). In the Down condition, NF signal values decreased significantly as sessions progressed, with a large effect size (session 1 > session 3, *z* = 3.829, *p* < 0.001, *r* = 0.629; session 1 > session 5, *z* = 4.619, *p* < 0.001, *r* = 0.759; session 3 > session 5, *z* = 3.547, *p* < 0.001, *r* = 0.575).

Negative correlations were observed between the performance changes in the recognition task and the NF signal changes. However, the statistical analysis revealed no significant correlations with the NF signal values for task 1 accuracy (*r* = −0.10, *p* = 0.95), recall (*r* = −0.10, *p* = 0.87), and task 2 accuracy (*r* = −0.87, *p* = 0.05). Furthermore, no significant correlations were found with other physiological data for task 1 accuracy (LTR, *r* = 0.30, *p* = 0.68; RIA, *r* = 0.10, *p* = 0.95), recall (LTR, *r* = 0.05, *p* = 0.93; RIA, *r* = 0.36, *p* = 0.55), and task 2 accuracy (LTR, *r* = 0.10, *p* = 0.87; RIA, *r* = −0.53, *p* = 0.36).

#### 3.2.4. NF Signal Transition within Sessions (P02)

For the first and final sessions, the within-session transitions of NF signal values were compared between Up and Down conditions (Figure 15). In the first session, both the Up and Down conditions exhibited higher values throughout almost all trials compared to trial 1. The Up condition had a greater number of trials surpassing trial 1 compared to the Down condition. On the other hand, during the final session, the NF signal value mostly exceeded that of trial 1 in the Up condition, while remaining lower than trial 1 in the Down condition across most trials.

#### 3.2.5. Low Theta Power Ratio (LTR) Changes across Sessions (P02)

The baseline LTR_1_ varied by a few percentage points between sessions: 5.65% in session 1, 6.04% in session 2, 7.23% in session 3, 10.02% in session 5, and 6.06% in session 6. A comparison between conditions was performed for LTR_i_ values from trial 2 onwards, similar to the NF signal analysis (Figure 16). When comparing Up and Down conditions for the same session number, no significant differences were observed in either the first session (*z* = 1.216, *p* = 0.233, *r* = 0.200) or the final session (*z* = −0.650, *p* = 0.531, *r* = 0.108). In the Up condition, as sessions progressed, there was a significant decrease in LTR_i_ values, with a large effect size (*z* = 3.565, *p* < 0.001, *r* = 0.594). However, there were no significant changes between sessions in the Down condition (session 1 vs. 3, *z* = 1.793, *p* = 0.075, *r* = 0.295; session 1 vs. 5, *z* = 0.881, *p* = 0.391, *r* = 0.145; session 3 vs. 5, *z* = −0.803, *p* = 0.435, *r* = 0.130).

#### 3.2.6. Frequency Band Power Changes across Sessions (P02)

The transitions between sessions of total power, delta power, and low theta power are presented in Figure 17. It is important to note that the vertical scale of the low theta power graph differs from the graphs of other frequency band powers. Delta power was consistently higher than low theta power throughout the sessions. When comparing the trends in NF signal values with each frequency band power, a negative correlation trend can be observed with all frequency bands, particularly with low theta power. Furthermore, the patterns of low theta power changes exhibited similarities to the performance changes in task 2. 

#### 3.2.7. Time–Frequency Maps for Correct and Error Trials (P02)

The time–frequency maps for Correct and Error trials are shown in Figure 18a,b. Both maps exhibited intermittent power increases in the theta, alpha, and high beta bands. It was observed that Correct trials displayed stronger and sustained power increases (Figure 18c). 

#### 3.2.8. Time–Frequency Maps for Up/Down and First/Final Sessions (P02)

The time–frequency maps for the Up and Down conditions, as well as the first and final sessions, along with the difference maps between sessions, are presented in Figure 19. In comparison between the Up and Down conditions, sustained power increases in the gamma band were observed for the Down condition compared to the Up condition, both in the first and final sessions. In the first session, a slight power increase from the low theta to beta frequency bands was observed for the Up condition compared to the Down condition. However, a significantly large cluster indicating a power decrease in the delta frequency band for the Up condition compared to the Down condition spread from the early to the mid-encoding period. In the final session, the Down condition exhibited a power increase compared to the Up condition across all frequency bands, starting before the presentation of the first-word stimulus. Additionally, a significantly large cluster indicating increased powers from the low theta to alpha frequency bands was observed for the Down condition compared to the Up condition during the early-encoding period. When comparing the first and final sessions, regardless of the Up or Down conditions, a power increase from the low theta to low gamma frequency bands was observed in the final session compared to the first session. In the Up condition, a significantly large cluster indicating a power increase was observed for the final session compared to the first session during the late-encoding period, spanning from the low theta to alpha frequency bands. Similarly, in the Down condition, a significantly large cluster indicating a power increase was observed for the final session compared to the first session, spreading from the early to late-encoding periods, encompassing the low theta to alpha frequency bands, and extending to the gamma frequency band during specific time periods.

### 3.3. Results for P03

P03 completed seven sessions of memory NF training over a span of four days. A seizure occurred on the day before Session 1, but the symptoms did not persist, and there were no residual symptoms during the training. For the occurrence rate of interictal activity, Session 4 exhibited a significantly higher rate than Session 2 (*z* = 3.030, *p* = 0.002, *r* = 0.479). After the series of NF training, P03 underwent right selective hippocampal-amygdala resection surgery, which resulted in the disappearance of seizures. In the post-surgery WMS-R assessment, the verbal memory index score was 92, showing a small change from the preoperative score of 91, indicating that verbal memory function was not impaired.

Throughout all sessions, P03 focused on memorization during the encoding period, and reports indicated that P03 did not discover the strategy to make the supporting team win until the end. There was a report of slight distraction after session 4. For session 5, it was observed that artificial noise with high amplitude was mixed during trial 1, making it difficult to adjust noise reduction using ASR. Since the NF signal values used trial 1’s LTR as the baseline, session 5 was excluded from the analysis.

#### 3.3.1. Intracranial Electrodes Used for Memory NF (P03)

In P03, subdural electrodes (Unique Medical, Tokyo, Japan) were implanted in the left MTL. Four platinum electrodes (3.0 mm in diameter) were laterally placed at 10 mm intervals (centre to centre) along the left parahippocampal gyrus, and intracranial EEG was measured and used for NF (Figure 20). A reference electrode placed on the dural side of the left temporoparietal lobe was used for P03.

#### 3.3.2. Performance Changes in the Recognition Task (P03)

The session-to-session changes of the three scores for the recognition task are shown in Figure 21. The performance of the task 1, for both accuracy and recall, decreased from session 1 to 2 and then increased. In contrast, the accuracy of the task 2 decreased from sessions 1 to 3 and then increased from sessions 4 to 6.

#### 3.3.3. NF Signal Changes across Sessions (P03)

The session-to-session changes in NF signal values (LTR_i/1_) are presented in Figure 22. The medians of the NF signal values were higher for Up compared to the baseline (LTR_1/1_) in session 1 (*Mdn* = 115.29, *n* = 19, *z* = 2.140, *p* = 0.032, *r* = 0.491), session 3 (*Mdn* = 103.34, *n* = 19, *z* = 0.850, *p* = 0.396, *r* = 0.195), and session 7 (*Mdn* = 101.26, *n* = 19, *z* = 0.295, *p* = 0.768, *r* = 0.068), although not significantly. However, in session 2 (*Mdn* = 167.96, *n* = 19, *z* = −4.621, *p* < 0.001, *r* = 1.060) and session 4 (*Mdn* = 140.12, *n* = 19, *z* = −4.621, *p* < 0.001, *r* = 1.060) for the Down condition, the NF signal values were significantly higher compared to the baseline, with a large effect size. In session 6 (*Mdn* = 94.43, *n* = 19, *z* = −0.850, *p* = 0.396, *r* = 0.195), the median of the NF signal values for Down was lower than the baseline, but not significantly. When comparing Up and Down conditions for the same session number, Down showed significantly lower values than Up in the first session (*z* = −4.744, *p* < 0.001, *r* = 0.770), as well as in the second session (*z* = −4.160, *p* < 0.001, *r* = 0.675), with a large effect size. However, in the final session, there was no significant difference in NF signal values between Up and Down (*z* = 0.336, *p* = 0.751, *r* = 0.054). In the Up condition, the NF signal values were significantly lower at the final session (session 7) than at the first session (session 1), with a moderate effect size (*z* = −2.409, *p* = 0.015, *r* = 0.391). There were no significant changes in NF signal values between other sessions for Up condition (session 1 vs. 3, *z* = 1.036, *p* = 0.311, *r* = 0.168; session 3 vs. 7, *z* = −1.445, *p* = 0.154, *r* = 0.234). In the Down condition, NF signal values decreased significantly as sessions progressed, with a large effect size (session 2 vs. 4, *z* = −3.606, *p* < 0.001, *r* = 0.585; session 2 vs. 6, *z* = −5.270, *p* < 0.001, *r* = 0.855; session 4 vs. 6, *z* = −5.065, *p* < 0.001, *r* = 0.822).

There were no significant correlations with the NF signal values for task 1 accuracy (*r* = −0.70, *p* = 0.12), recall (*r* = −0.72, *p* = 0.11), and task 2 accuracy (*r* = 0.03, *p* = 1.00). Furthermore, no significant correlations were found with other physiological data for task 1 accuracy (LTR, *r* = −0.15, *p* = 0.77; RIA, *r* = 0.39, *p* = 0.44), recall (LTR, *r* = 0.48, *p* = 0.34; RIA, *r* = 0.84, *p* = 0.04 > 0.05/3), and task 2 accuracy (LTR, *r* = −0.26, *p* = 0.66; RIA, *r* = 0.09, *p* = 0.92).

#### 3.3.4. NF Signal Transition within Sessions (P03)

For the first and final sessions, the within-session transitions of NF signal values were compared between Up and Down conditions (Figure 23). In the first session, the Up condition initially showed a lower trend but gradually increased above trial 1 as the session progressed. On the other hand, the Down condition consistently exhibited higher values than trial 1 throughout all trials, showing higher values than the Up condition. However, in the final session, neither the Up nor Down condition showed significant increases nor decreases from trial 1.

#### 3.3.5. Low Theta Power Ratio (LTR) Changes across Sessions (P03)

The baseline LTR_1_ varied between sessions: 25.01% in session 1, 20.69% in session 2, 28.18% in session 3, 25.01% in session 4, 34.81% in session 6, and 34.41% in session 7, with late sessions being larger than early and mid sessions. A comparison between conditions was performed for LTRi values from trial 2 onwards, similar to the NF signal values (Figure 24). The comparison of conditions for the same session number revealed that the Down condition had significantly higher LTR values than the Up condition in the first session (*z* = −2.817, *p* = 0.004, *r* = 0.457) and second session (*z* = −2.701, *p* = 0.006, *r* = 0.438), with a moderate effect size. In the Up condition, there were no significant differences between sessions (session 1 vs. 3, *z* = −0.219, *p* = 0.840, *r* = 0.036; session 1 vs. 7, *z* = −1.825, *p* = 0.070, *r* = 0.296; session 3 vs. 7, *z* = −1.562, *p* = 0.123, *r* = 0.253). Similarly, in the Down condition, there were no significant differences between sessions (session 2 vs. 4, *z* = −0.511, *p* = 0.624, *r* = 0.083; session 2 vs. 6, *z* = 0.423, *p* = 0.686, *r* = 0.069; session 4 vs. 6, *z* = 1.036, *p* = 0.311, *r* = 0.168).

#### 3.3.6. Frequency Band Power Changes across Sessions (P03)

The changes in total power, delta power, and low theta power across sessions are illustrated in Figure 25. In the early sessions, delta power showed a more prominent difference than low theta power between Up and Down conditions (Up > Down). Negative correlations were observed between all band powers compared to the changes in NF signal values.

#### 3.3.7. Time–Frequency Maps of Correct/Error Trials during Encoding Period (P03)

Time–frequency maps for Correct and Error trials are shown in Figure 26a,b. Both maps display intermittent power increases in the theta and alpha bands, with Error trials exhibiting stronger and more sustained power (Figure 26c).

#### 3.3.8. Time–Frequency Maps for Up/Down and First/Final Sessions (P03)

The time–frequency maps for the Up and Down conditions, as well as the first and final sessions, along with the difference maps between sessions, are presented in Figure 27. In comparison between the Up and Down conditions, both in the first and final sessions, the Up condition showed more continuous power increases across a wide frequency range from delta to gamma. In the first session, a significantly large cluster indicating a power increase, primarily centred around the delta and theta bands, extended throughout the entire encoding period, expanding to the gamma band during specific time periods. When comparing the first and final sessions, both Up and Down conditions showed power increases from the low theta to beta bands in the final sessions compared to the first session. In the Up condition, a significantly large cluster indicating a power increase was observed during the early-encoding period, covering the low theta to beta band. On the other hand, in the Down condition, a significantly large cluster indicating a power increase was observed throughout the entire encoding period, covering the low theta to alpha bands, expanding to the beta band during specific time periods.

## 4. Discussion

In this study, we conducted bidirectional NF training in three patients with drug-resistant epilepsy to modulate the LTR of the MTL. We observed an increased difference in NF signals (Up−Down) as sessions progressed. While the NF signal changed significantly only in one direction for all participants, the opposite-direction NF acted as an intra-subject control, allowing us to exclude potential confounders, such as motivation, visual modality, and eye movement, whose influences were nearly identical across conditions.

Changes in NF signal values in only one direction can be attributed to the effect of anterograde interference observed in motor and perceptual learning [72,73]. This interference reflects how the memory of one task influences the learning of a subsequent task. The influence of the subsequent task on the memory of the prior one is termed retrograde interference and has fewer effects than anterograde interference [73]. Previous studies using fMRI-based bidirectional NF reported that anterograde interference could impact the latter half NF sessions [74]. This might explain discrepancies between the present and our prior study, where we performed NF using intracranial electrodes and observed increased theta power in one participant [24]. In the present study, despite the same electrode location for P02, we found NF signal values decreased over sessions for both Down and Up conditions. This may indicate that the memories of Down NF influenced Up NF learning, though identifying a precise cause is challenging due to various differences between the two studies, such as the number of sessions for Up condition, NF signal, and NF modality.

We found no correlation between the transitions of memory task scores and NF signal values. This lack of correlation may stem from the within-subject design causing the two types of NF to interfere with each other [74,75]. Another factor could be the lack of a clear mental strategy for NF, leading one participant (P01) to randomly answer the recognition task, thereby significantly reducing accuracy in task 1. Providing a clear mental strategy at the outset might prevent unforeseen strategies and enhance the success of NF. While the relationship between the provision of a mental strategy and the outcomes of NF is under investigation [76], systematic studies are scarce [77], and there are no established standards for providing strategies [78]. Based on instructional design research, effective feedback requires well-defined goals [79]. Thus, providing participants with clear objectives and mental strategies could enhance NF training by minimising trial and error [76]. However, a mental strategy might create a dual-task situation with the memory task, dividing attention and indirectly affecting the performance of the task [80,81]. Insufficient trials per session may have also affected the lack of correlation, but adding more trials was impractical due to concerns about patient burden. Future research could consider introducing memory tasks separate from memory NF for an independent evaluation of memory function.

In the time–frequency analysis, we found intermittent increases in theta and alpha power in both correct and error trials regarding presented word order. Past research has suggested an association between theta oscillations and memory [44,82], with similar increases reported in scalp EEG [41,47,83,84,85], MEG [42,86], and iEEG studies [43,45,46,87], corroborating our study results. Theta and alpha power are also suggested to be linked with memory retention [84,88]. Studies have shown that these powers in the MTL increase with memory load [89], and alpha activity rises during word sequences retention [90]. In our study, the memory load was heavier, as participants were questioned on word order in task 2. This might have caused the neuronal activity associated with word retention to be reflected more in the encoding period than in studies only querying word presence. In comparing correct and error recognitions of word order, two participants (P01, P03) exhibited a relative decrease in theta power during successful encoding, aligning with numerous intracranial studies [45,54,58,91,92,93]. Notably for P01, a relative increase in low theta power before stimulus presentation and a decrease during stimulus presentation were observed in relation to successful encoding. Previous studies have reported an increase in low theta power before stimulus presentation, associating it with the preparatory process for encoding [43,87,94]. For P01, this preparatory state might have contributed to the recognition task’s success. On the other hand, P02 exhibited intermittent increases in theta and alpha power, linked to successful encoding, along with an increase in the beta band. Most conventional studies reporting a decrease in power in these bands when questioning the presence of words [52,54,57,92,93,95,96] may seem contradictory to P02’s results at first glance. In this study, however, we based the correct and error encoding on the recognition of presented word order, analysing 13 s including before and after the presented five words. Thus, compared to traditional studies, this investigation might better reflect neural activity related to word retention and order-memory. As P02 scored lower on tasks than other participants, the memory load or difficulty may have been relatively high. Since theta and alpha power during memory retention increases with load [84,89,90], P02 may have been more affected by memory retention on the scores of tasks than other participants, leading to different results. The increase in beta power in the MTL during successful encoding is thought to relate to sensory processing [45,87]. Additionally, increased alpha and beta band activity before the presentation of the encoding word have been reported in the neocortex and MTL [43,87], suggesting associations with predictive and attention processes [48,54], and inhibitory top-down control of sensory processing [43,45,54]. In P02, these cognitive functions may have more significantly affected successful encoding than in other participants. P02 even commented on employing mental strategies, such as concentration adjustment during encoding, supporting the suggestion of previous studies that volitional control can regulate a brain network involving the hippocampus, dorsolateral prefrontal cortex, and cerebellum, thereby benefiting memory function [97].

In the time–frequency analysis, no significant changes in the difference of gamma power between the Up and Down conditions were observed when comparing the first and final sessions. Specifically, regardless of the first or final sessions, two participants (P01 and P03) showed intermittent increases in gamma power in the Up condition compared to the Down condition, while P02 showed the opposite. The small change before and after training might stem from the exclusion of the gamma band in the calculation of the NF signal values, possibly diminishing the NF effect compared to other bands. Increased gamma band activity during encoding has been reported [46,48,54,55,70,93,98], and is thought to reflect an increase in attentional resources [48,70] and the association of stimuli with spatiotemporal context, thereby aiding the formation of episodic memories [56,95]. Regarding alpha and beta power, two participants (P02 and P03) showed an increase in the final versus first sessions, independent of the Up/Down condition. A similar increase in low theta power was observed, yet with a decrease in NF signal values across both conditions, suggesting that the alpha and beta bands’ activity had a relatively high influence. Given the association of alpha and beta oscillations with encoding-related predictive and attentional processes, and inhibitory top-down control [43,45,48,54], familiarity with the tasks across sessions may have influenced these cognitive functions. On the other hand, for P01, although the relationship of alpha and beta power between conditions was firstly Up < Down, the subsequent sessions led to an increase in the Up condition and a decrease in the Down condition, resulting in Up > Down in the final session. This shift implies that the NF training, targeting theta power adjustment, could have influenced predictive and attentional processes and inhibitory top-down control, thereby possibly leading to an indirect adjustment of alpha and beta power. 

Memory and attention play a central role as fundamental processes in cognitive processing, and are closely related [99,100,101,102]. For example, the availability of attentional resources and selective attention are crucial for successful memory encoding [100]. The availability of attention resources is thought to be reflected by the deactivation of the Default Mode Network (DMN), including the temporoparietal regions and the posterior cingulate gyrus [103], and it has been shown that NF towards the deactivation of the DMN improves sustained attention [104]. Selective attention has also been shown to improve with NF of the sensorimotor rhythm (SMR) [105]. Moreover, NF of the SMR has been reported to enhance working memory [106]. In the same way that attention affects memory, memory also affects attention. The hippocampus, for example, receives subcortical inputs from neurotransmitter systems involved in attention, such as the cholinergic and dopaminergic systems. Additionally, the hippocampus has neuroanatomical connections with brain regions that play a role in orienting attention, such as the posterior parietal cortex and the thalamus, as well as with regions that play a role in executive attention, including the anterior cingulate cortex, thereby modulating attentional processes [102]. In this study, we have found the potential that the low theta power of the MTL indirectly adjusts higher frequency bands, affecting the mnestic processes, including attention resources, predictive and attention processes. Despite its limitations as a pilot study, requiring further additional evaluation, the present study suggest that NF using iEEG of MTL is useful in making causal inferences about the interactions between memory and attention, and in deepening our understanding of the broad dynamics of cognitive processing.

On the other hand, recent research has reported an intriguing result that cognitive training, not NF, has improved working memory [107]. This study extracted alpha oscillatory activity during the visual object storage period from a total of eight electrodes placed over a wide range of the scalp (central, parietal, and temporoparietal regions) and in the ear, and suppressed it by performing NF with mental calculation and mental imagery. Their approach and our study are similar in that both conduct NF while imposing memory tasks. Still, they differ in the targeted memory type (visual or verbal), process (encoding or storage), brain activity (regions, frequencies), and NF implementation methods (providing clear mental strategies, real-time or intermittent, reward type, etc.). These differences could be factors influencing the results. In particular, targeting specific brain regions associated with cognitive functions is vital for reducing irrelevant noise, enhancing signal accuracy and specificity, and succeeding in NF. In fact, NF studies that target frontal-midline theta, which contributes to memory processing, such as hippocampal theta rhythms [108], have reported improvements in episodic and semantic memory, and acceleration and reinforcement of memory consolidation [30,31,33,36,37]. In our research, we focused on the MTL as a target and performed bidirectional NF to exclude potential confounding factors. Still, we cannot completely exclude the effects of cognitive training. For future research directions, it would be important to evaluate the pure effects of NF by comparing conditions in which NF training is conducted with those in which only cognitive training is performed.

Next, we will proceed with the interpretation of the results specific to each individual participant.

### 4.1. Discussion of the Results of P01

In the case of P01, it is inferred that the left side was dominant for verbal memory function, seeing as verbal memory function declined after surgery. Unlike other participants, P01 received NF for the right MTL, more closely linked to visual memory function, rather than the left MTL, which is closely related to verbal memory function. In this study, despite the use of a standard verbal memory task, NF training led to differences in the right MTL’s activity and performance on task 2 between the Up and Down conditions. The possible reasons for this could be: 1. the visual presentation of words on a monitor might have evaluated not only verbal encoding function but also visual encoding function; 2. the right side might also be responsible for verbal encoding, not solely the left side. To address 1, using stimuli that are less likely to involve visual processing, such as auditory stimuli, could more precisely assess NF’s effect solely on verbal encoding function in future studies. Regarding 2, based on the elapsed years since the onset of epilepsy and the presence of hippocampal sclerosis in the left MTL, it can be inferred that verbal memory function may have been partially reorganized towards the right MTL, potentially serving as an alternative or complementary role of the left MTL. Regardless, as evidenced by the observed similarity in the trends of NF signal values and performance on memory tasks, it is possible that NF not only adjusted neural activity but also memory function.

Considering that there was a difference in total power, which is the denominator of NF signal values, between conditions (Up > Down), it is plausible that the trend in NF signal values was largely contributed to by the trend in low theta power. Moreover, the similarity with task 2’s accuracy transition indicates that low theta power is likely to have played a crucial role in encoding.

In the Up condition, the NF signal values significantly increased from the first to the final session, and in the intra-session trends of NF signal values a steeper rise from trial 1 was observed during the final session compared to the first session. Time–frequency map comparisons confirmed the intermittent increase in theta power in the final session compared to the first. These results might imply a cumulative NF effect in the Up condition. On the other hand, the adjustment in theta power could be related to reward expectation. The relationship between reward and theta activity has been suggested in rodent studies [109], and human studies have indicated that reward motivation is associated with successful memory encoding [110,111]. One study reported an increase in theta power just before word presentation associated with successful encoding under high reward conditions [112]. Although not mentioned by the authors, Figure 3C of this paper shows a visible increase in theta power immediately after word presentation associated with successful encoding under low reward conditions. In P01’s session 3, due to a system error, NF continued to make it difficult for the red team to win, which might have created a low reward condition unexpectedly, potentially inducing an increase in theta power.

### 4.2. Discussion on the Results of P02

The transition in NF signal values and frequency band powers across sessions showed a negative correlation trend. This suggests that the transition in NF signal values was more influenced by other frequency band power rather than by low theta power. Specifically, delta power had a higher value than low theta power, potentially exerting a greater influence on total power. Consequently, the transition in NF signal values may have strongly reflected the change in delta power.

We observed a negative correlation trend between the scores of the recognition task and NF signal values. However, given the similarity between task performance and low theta power, it is plausible that in P02, low theta power played a crucial role in encoding. Despite our intention to modulate neural activity contributing to memory encoding with the Up/Down conditions, it is possible that the NF had resulted in the opposite direction of modulation.

In the Up condition, there was a trend in decline in NF signal values and LTR_i_ values over sessions. This trend could be attributed to the rise in total power, especially the rise in delta power, outweighing the increase in low theta power. The same declining trend of NF signal values was observed in the Down condition, along with a rising trend in power across frequency bands (total power, delta power, low theta power. However, no significant differences in LTR_i_ values were observed between sessions, possibly because low theta power increased as much as total power. The strong upward trend in low theta power in Down compared to Up may be the result of an unintentional adjustment of low theta power in the opposite direction in both conditions. One direct reason for the decline in NF signal values in Down could be the rising LTR_1_ values, potentially a carryover effect from the previous Up session.

P02 reported in the later sessions that concentrating to memorise led to the Red team winning, while memorising with relaxation led to the White team winning. This could indicate a difference in load during memory retention. Some studies have reported increased theta activity in the frontal midline area during tasks requiring focused attention or high-load memory tasks [113,114,115]. The hippocampus, strongly connecting with the prefrontal cortex [116,117], is suggested to contribute to cortical theta activity through their interaction [40]. Furthermore, one study reported an increased MTL’s theta power during memory retention with increasing memory load [89]. Considering these factors, when concentration was required for memory, theta power is likely to have increased, boosting NF signal values and leading the Red team to win; in relaxation, it likely decreased, causing the White team to win. In the first sessions, regardless of conditions, the NF signal values were higher than the baseline of trial 1. This might be due to an increase in theta power, as P02 still needed to focus on memorising. Additionally, as the scores in task 2 were lower in the first sessions compared to other sessions, it can be inferred that the memory task imposed a high load during these first sessions.

### 4.3. Discussion of the Results of P03

Interestingly, in the first and intermediate sessions of NF training, the NF signal values and LTR_i_ were higher in Down than in Up. Given the negative correlation trend between the transitions in NF signal values and transitions in frequency band powers, it can be inferred that other frequency bands had more influence on the transitions in NF signal values than low theta power did. In particular, delta power exhibited more significant differences between conditions (Up > Down) in the first and intermediate sessions than low theta power did, with the magnitude of this power difference being similar to that in the total band. We also observed intermittently lower delta power in Down than in Up during the first session comparison of the time–frequency map between conditions. Considering these, we can infer that delta power significantly influenced LTR and NF signal values. Although we did not observe any similarities or negative correlations between the transitions of neural activity and the transitions of scores in the memory task, the differences between NF conditions were most apparent in the delta band. This might indicate that in P03, delta power is linked with memory encoding function, similar to or even more than low theta power. Some studies have reported modulation of delta power during encoding [45,52,53,96], with Lega et al. [45] highlighting its functional significance. They noted the similarity between oscillations below 4 Hz in the human MTL and memory-related theta oscillations observed in the hippocampus of animals. If delta power contributed more to encoding than low theta power in P03, then, given the observed negative correlation trend between NF signal values and delta power, we may have unintentionally led to NF that decreased delta power in Up and increased it in Down.

To investigate the causes of the differences in NF signal values and LTR_i_ (Up < Down) in the first and intermediate sessions, we examined the within-session transitions in delta power and low theta power, as shown in Figure 28.

In Up, delta power transitioned higher than low theta power, suggesting a significant impact on NF signal values. In Up, delta power remained high for the first few trials but later declined to levels akin to trial 1. In the first session of Up, NF signal values were lower than trial 1 for the first few trials but then rose to values similar to or higher than trial 1. These may imply that in P03, the red team’s loss in the first few trials in Up played as a negative reinforcement in NF learning, leading to delta power decline and changes in NF signal values. However, LTR_i_ and NF signal values were not elevated significantly beyond the baseline. This could be due to the loss of negative reinforcement when escaping from the losing situation and subsequent repetitions of the increase and decrease (down to the level of trial 1) of NF signal values, making it difficult for positive reinforcement to occur. Furthermore, the inability to observe an increase in LTR_i_ and NF signal values might be attributed to the potential retrograde effect of interposing the Down session between Up sessions.

On the other hand, in Down, delta power remained lower and more stable than in Up throughout the session, with NF signal values remaining high. This pattern could be interpreted as either the anterograde carryover effect of lowering delta power in the previous Up session or a decrease in motivation for NF due to continuous losses. However, an increasing trend in delta power across Down sessions may imply that the continuous losses served as negative reinforcers. In Down, as delta power increased, low theta power also increased, with no differences in LTR_i_ observed between sessions. Nevertheless, NF signal values declined with each Down session, possibly caused by the magnitude of LTR_1_, increasing across sessions, which might be a carryover effect from the previous Up session.

### 4.4. Limitation and Future Prospects

This study has several limitations. Firstly, we adopted the band power ratio rather than band power as NF signal values. Due to variations in frequency power between individuals and the possibility of diurnal and inter-day changes, we thought that using the band power and visualising it for the tug-of-war game would be challenging. We believed that adjusting the LTR would lead to adjustments in low theta power. However, the large fluctuations in delta power influenced changes in LTR in some participants. Thus, future studies could focus on specific frequency band power for NF to prevent the influence of other frequency activities. It may also be important to perform a frequency analysis beforehand to optimise the target frequency band individually. Nevertheless, having used the band power ratio as the NF signal allowed us to recognise the importance of other frequency bands in encoding, suggesting memory NF could potentially influence multiple frequency bands.

Secondly, we switched the direction of NF for each session for the conducting of bidirectional NF. Bidirectional NF is a powerful control condition for excluding potential confounding factors [75]. However, this resulted in having only 2–4 sessions per condition. It has been suggested that intermittent NF takes time to learn [118], and the limited number of sessions per condition might have influenced the results of our NF training. Besides, there might have been anterograde and retrograde effects that could interfere with NF learning. However, this also could suggest that the effects of memory NF could continue, even after some time. Future studies could conduct training for one condition first before switching to the other condition. This would enable us to assess the accumulation of new NF effects, even as the influences from previous NF condition remain. Conducting group comparisons by assigning participants to Up and Down conditions also excludes anterograde and retrograde effects and increases the number of sessions per condition.

Finally, we limited our observations to a qualitative statement of the relationship between the transition of neural activity and changes in memory task scores, where similarities and negative correlation trends were observed. The calculations of frequency band power were executed as exploratory analyses. Thus, we refrained from conducting correlation analyses and applying multiple comparison corrections, as this could have increased the risk of committing type II errors, leading to erroneous conclusions. Furthermore, it must be noted that this study had insufficient sessions available to stably evaluate the correlation. However, considering the burden on participants and the number of electrode placement days, securing the number of sessions required for a stable quantitative evaluation proved challenging.

## 5. Conclusions

In this study, we created an NF system to control the neural activity in the MTL bidirectionally (Up and Down) and evaluated its effects on neural activity and memory function after several days of training. Specifically, a memory NF system was developed that modulated neural activity using the low theta power ratio from intracranial electrodes during memory encoding.

Three patients with drug-resistant epilepsy underwent NF training, leading to an increased difference in the NF signal conditions (Up–Down) as sessions progressed. However, all participants were able to control neural activity only unidirectionally, not bidirectionally. We found that NF may influence memory retention and various cognitive functions related to memory, including prediction, attention processing, and inhibitory top-down control of sensory processing. Furthermore, we identified potential contributions of frequency bands below 2 Hz to encoding and observed similarities and a negative correlation tendency between neural activity and memory function transitions. 

Our research represents a crucial step in understanding NF’s application to memory and provides insights that could inform the creation of more effective memory enhancement strategies.

## Figures and Tables

**Figure 1 biomedicines-11-02262-f001:**
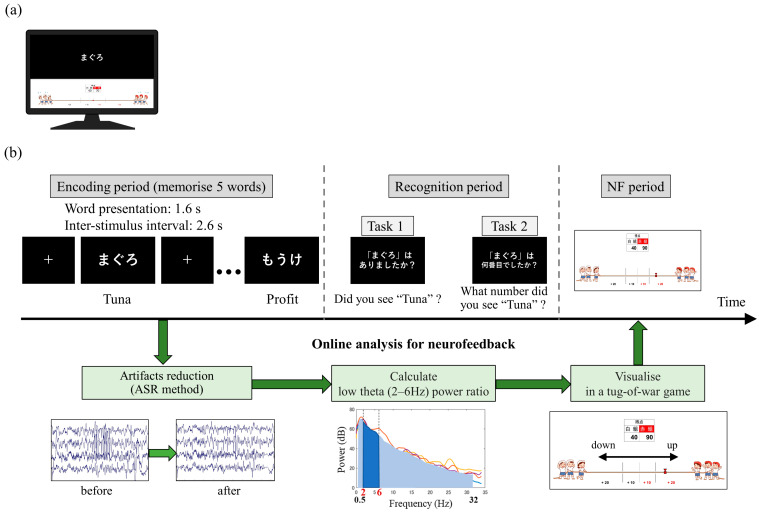
Memory neurofeedback paradigm. (**a**) Example of monitor screen during the encoding period. (**b**) One-trial sequence of memory neurofeedback and the process of online analysis. NF, neurofeedback; ASR, Automatic Subspace Reconstruction.

**Figure 2 biomedicines-11-02262-f002:**
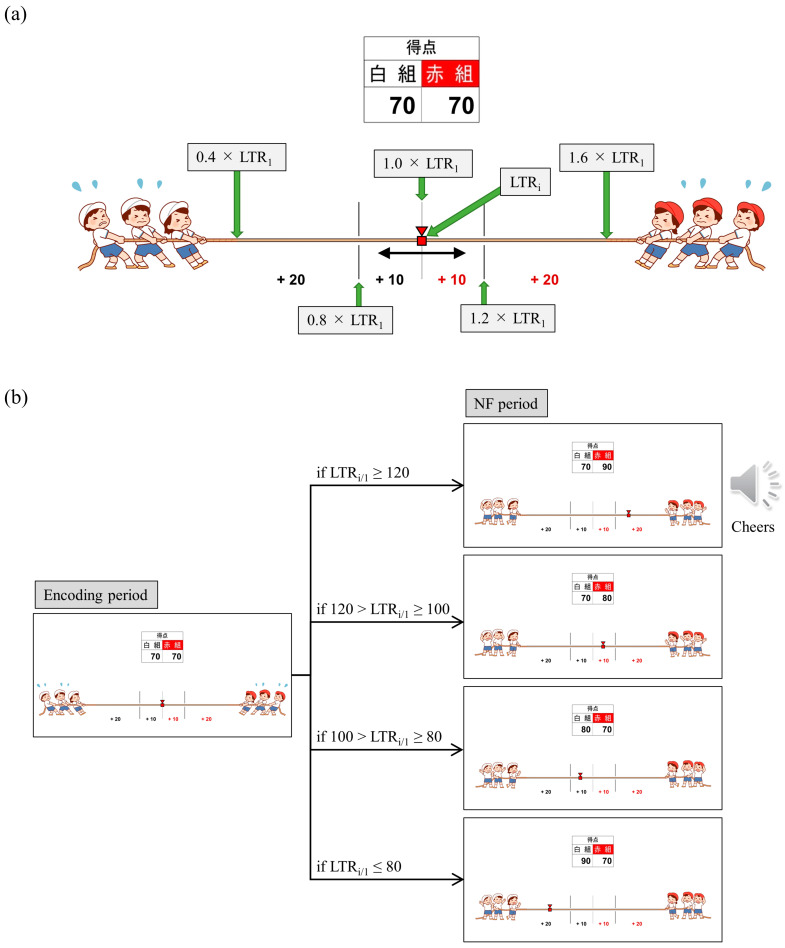
Neurofeedback visualization in tug-of-war. (**a**) Design of Neurofeedback screen. (**b**) Transition of the NF screen from the encoding period to the NF period in trial i of sessions rooting for the red team (Up condition). LTR, the average ratio of low theta band power; NF, neurofeedback.

**Figure 3 biomedicines-11-02262-f003:**
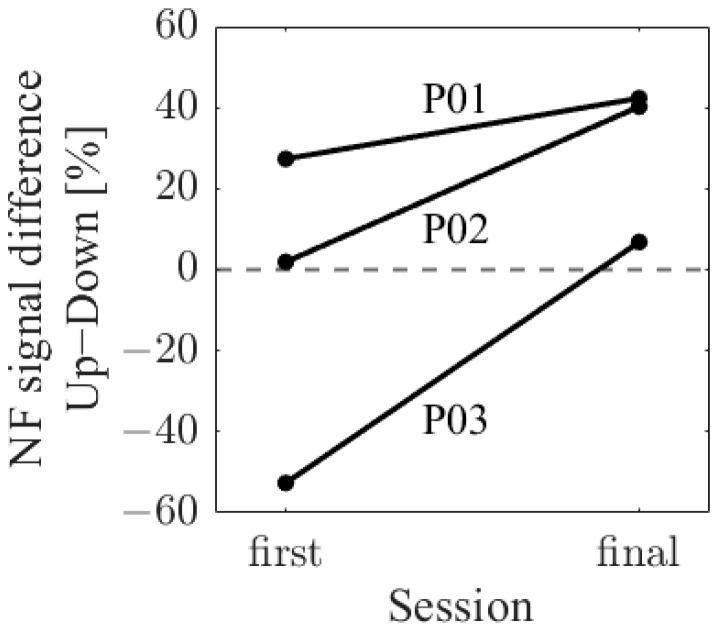
Changes in neurofeedback signal difference (Up–Down).

**Figure 4 biomedicines-11-02262-f004:**
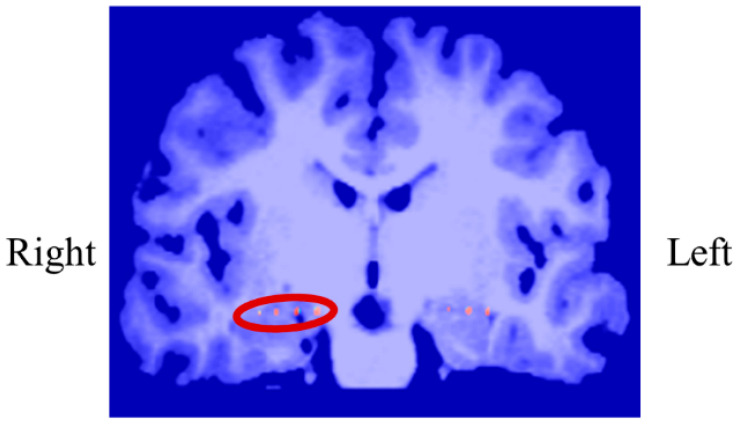
Electrodes of P01 used for neurofeedback.

**Figure 5 biomedicines-11-02262-f005:**
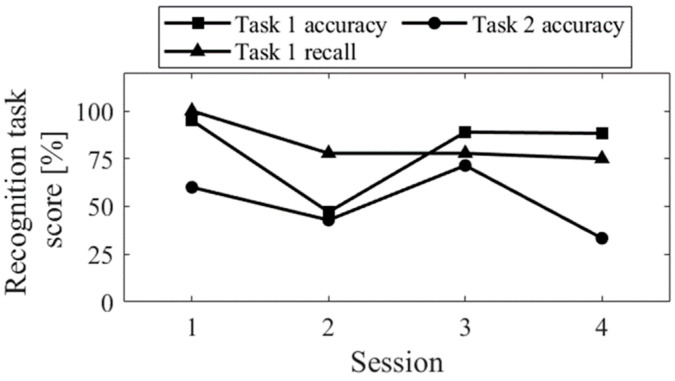
Changes in recognition task performance of P01 across sessions.

**Figure 6 biomedicines-11-02262-f006:**
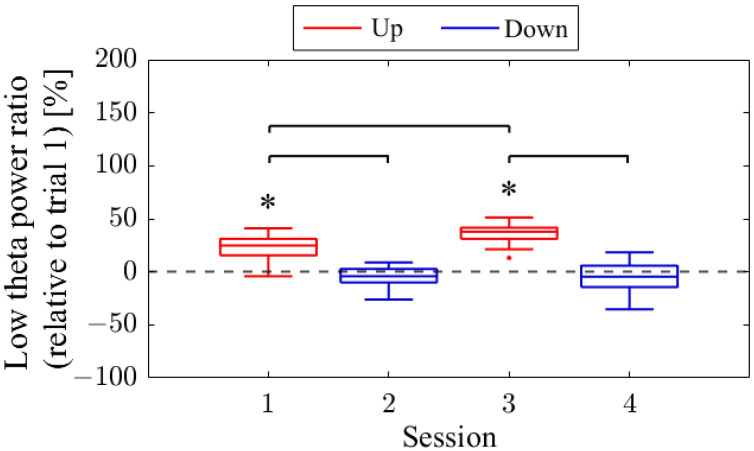
Changes in NF signal of P01 across sessions. * *p* < 0.05 compared to trial 1.

**Figure 7 biomedicines-11-02262-f007:**
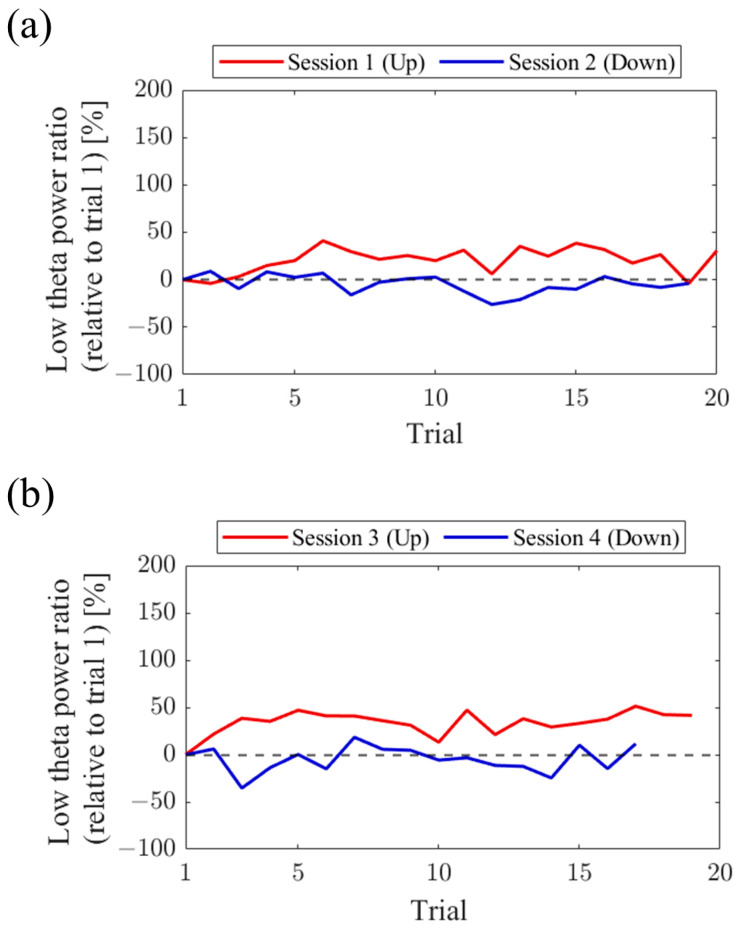
Changes in NF signal of P01 within session. (**a**) The first sessions. (**b**) The final sessions.

**Figure 8 biomedicines-11-02262-f008:**
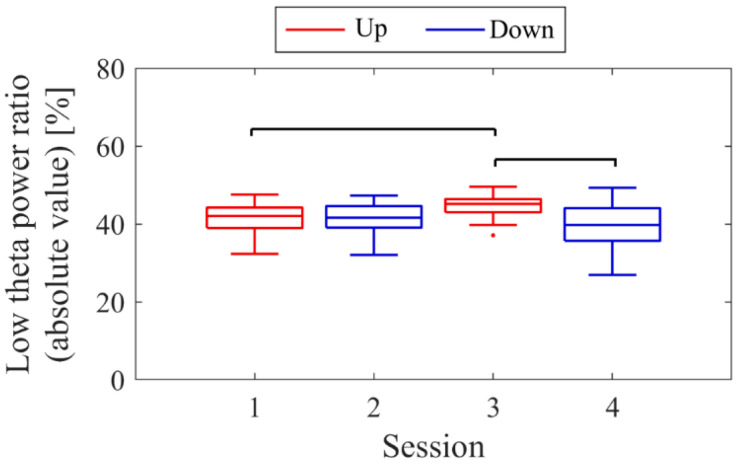
Changes in Low theta power ratio of P01 across sessions.

**Figure 9 biomedicines-11-02262-f009:**
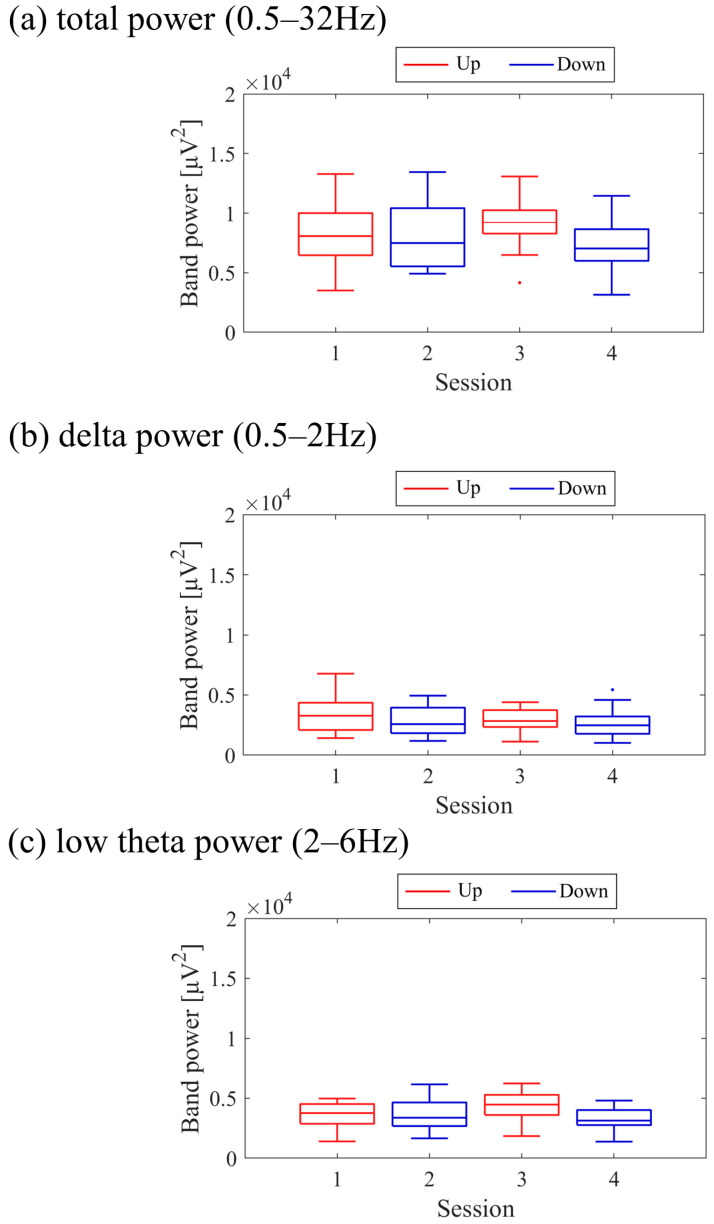
Changes in band power of P01 across sessions.

**Figure 10 biomedicines-11-02262-f010:**
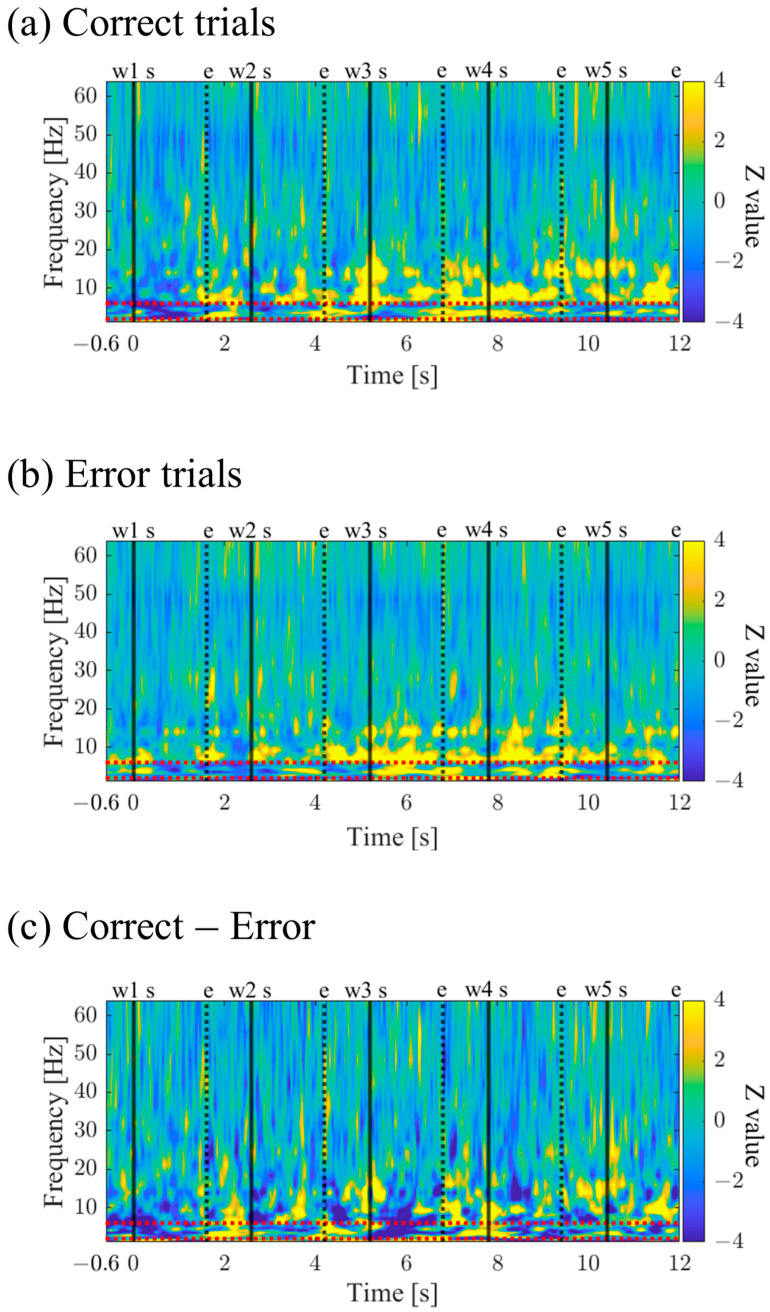
Time-frequency maps of P01 during memory encoding: (**a**) the correct trials, (**b**) the error trials, and (**c**) the difference (Correct trials–Error trials). Time–frequency spectral power was normalised to the baseline activity (−0.6 to −0.1 s), with time represented on the *x*-axis (−0.6 to 12 s, with the first stimulus presentation at 0) and frequency on the *y*-axis. The solid black vertical line labelled “ws” indicates the onset of each stimulus presentation, while the number between “w” and “s” denotes the number of the stimulus. The dotted black vertical line labelled “e” represents the offset of each stimulus presentation. The two red horizontal dotted lines represent the lower and upper boundaries of the low theta band (2 Hz and 6 Hz).

**Figure 11 biomedicines-11-02262-f011:**
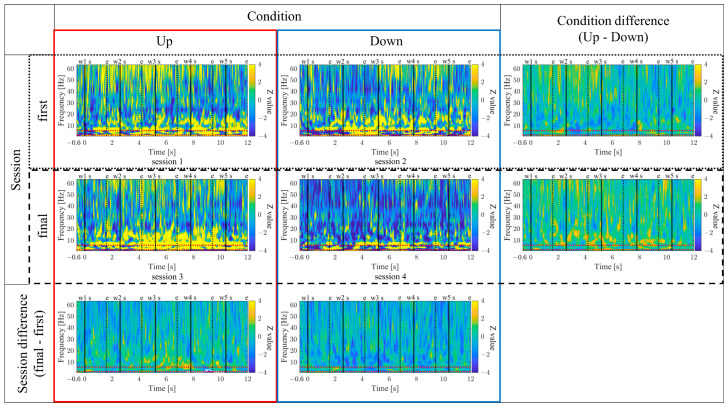
Time–frequency maps of P01 (Up/Down, the first/final sessions, and the difference). Figure caption omitted as it duplicates Figure 10’s caption. The area enclosed by the solid white line indicates clusters of significantly larger size.

**Figure 12 biomedicines-11-02262-f012:**
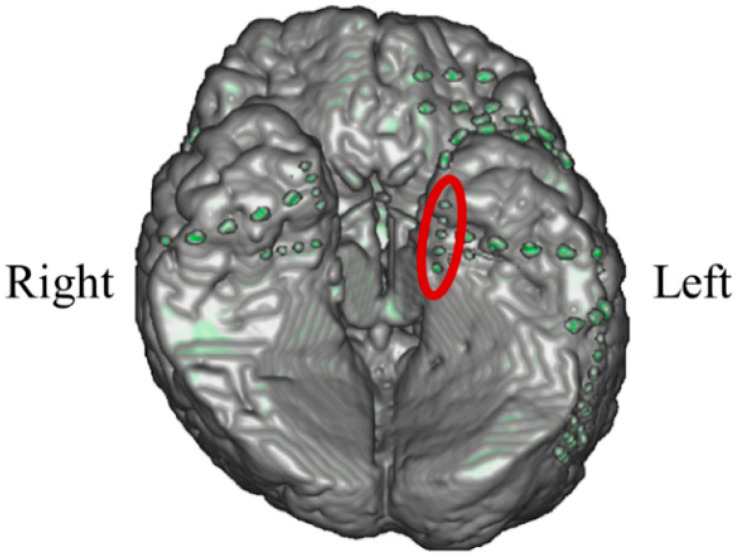
Electrodes of P02 used for neurofeedback.

**Figure 13 biomedicines-11-02262-f013:**
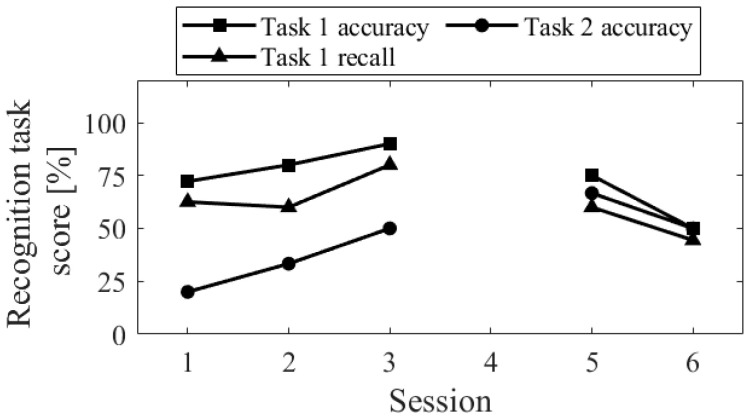
Changes in recognition task performance of P02 across sessions.

**Figure 14 biomedicines-11-02262-f014:**
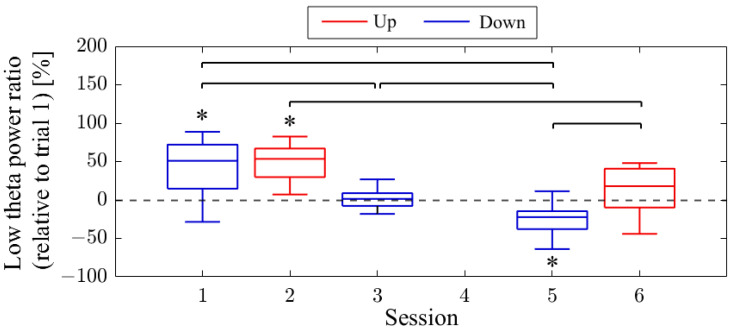
Changes in NF signal of P02 across sessions. * *p* < 0.05 compared to trial 1.

**Figure 15 biomedicines-11-02262-f015:**
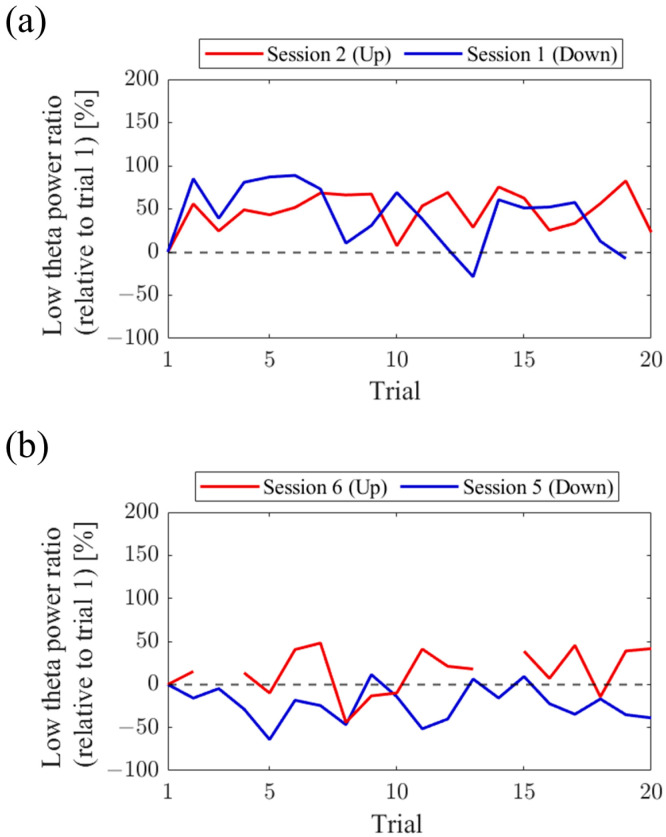
Changes in NF signal of P02 within session. (**a**) The first sessions. (**b**) The final sessions.

**Figure 16 biomedicines-11-02262-f016:**
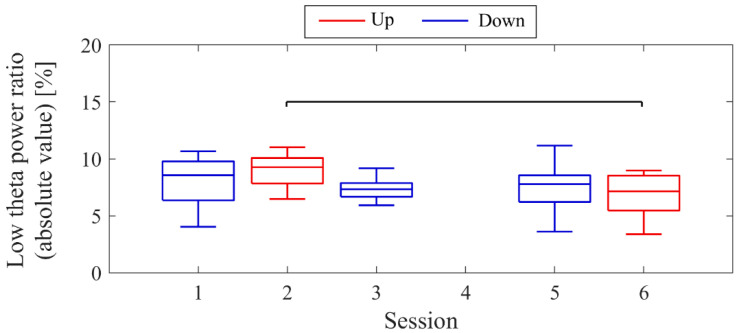
Changes in Low Theta Power Ratio of P02 across sessions.

**Figure 17 biomedicines-11-02262-f017:**
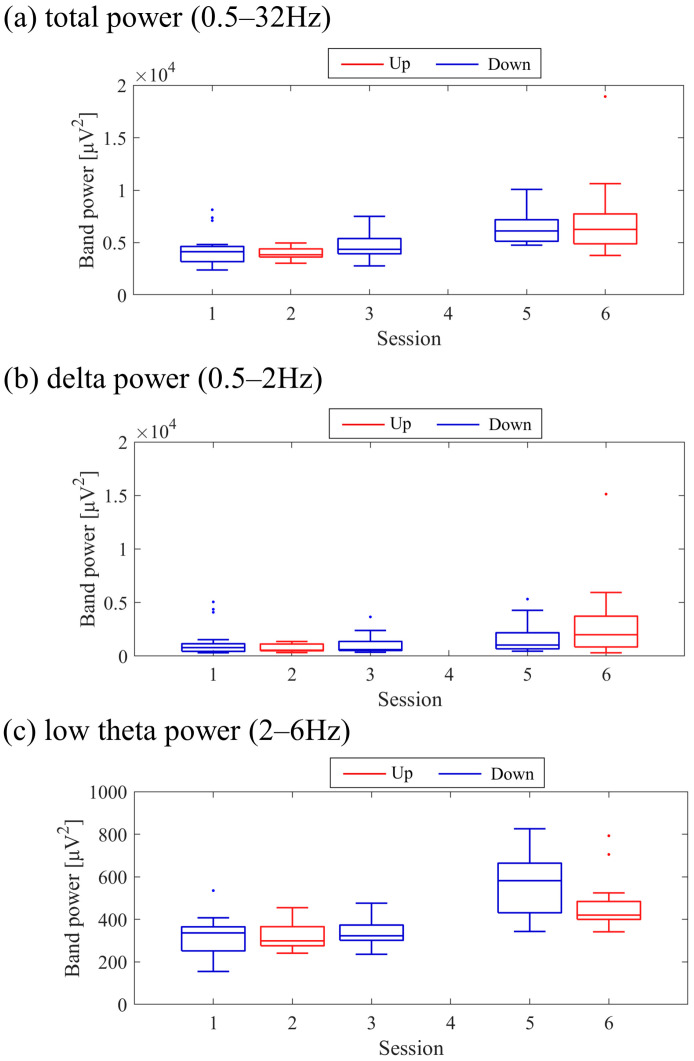
Changes in band power of P02 across sessions.

**Figure 18 biomedicines-11-02262-f018:**
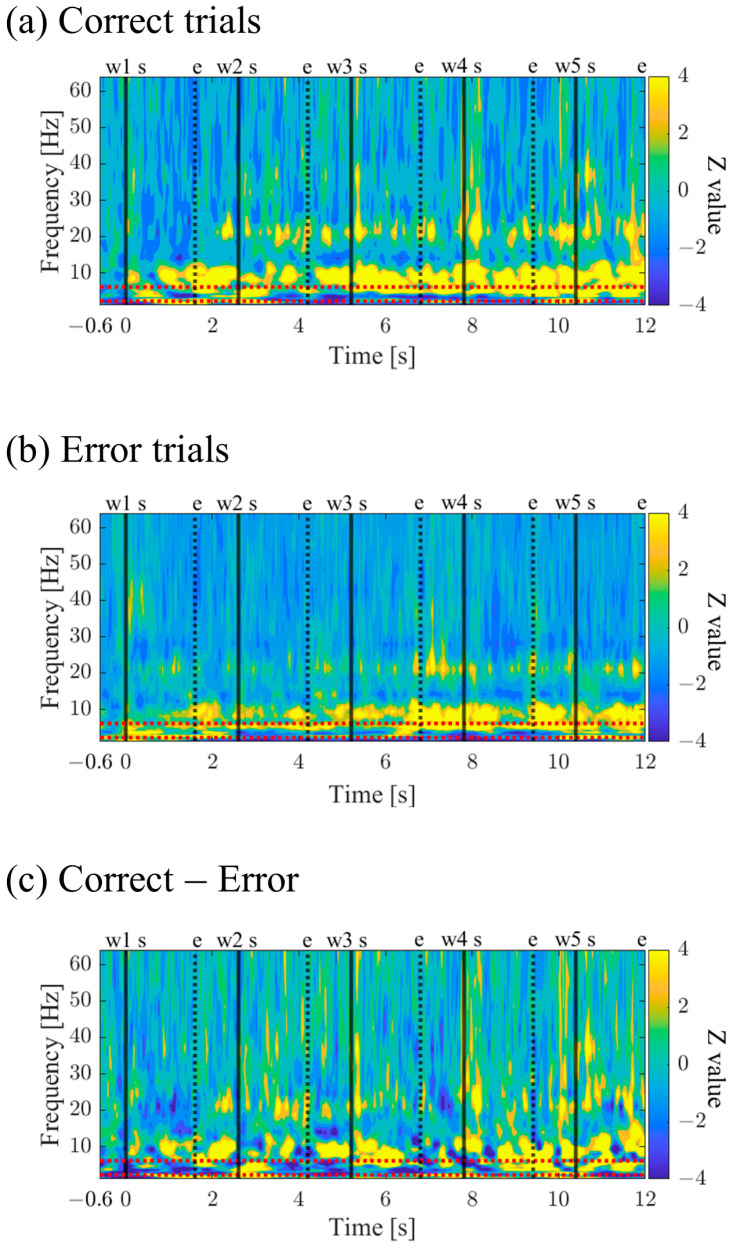
Time–frequency maps of P02 during memory encoding: (**a**) the correct trials, (**b**) the error trials, and (**c**) the difference (Correct trials–Error trials). Figure caption omitted as it duplicates Figure 10’s caption.

**Figure 19 biomedicines-11-02262-f019:**
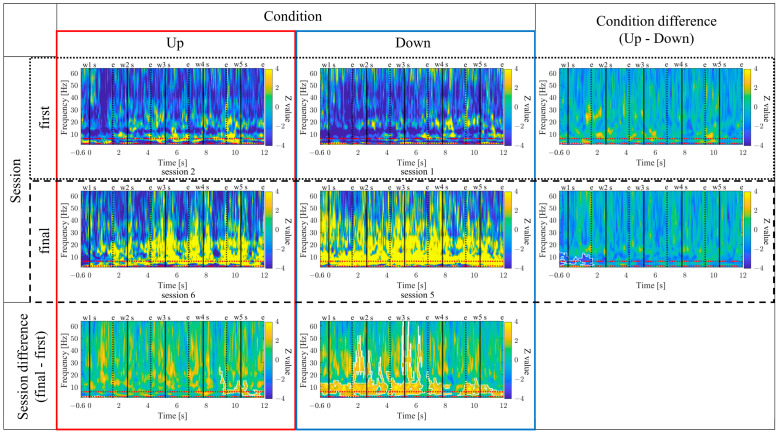
Time–frequency maps of P02 (Up/Down, the first/final sessions, and the difference). Figure caption is omitted as it duplicates Figure 10’s caption. The area enclosed by the solid white line indicates clusters of significantly larger size.

**Figure 20 biomedicines-11-02262-f020:**
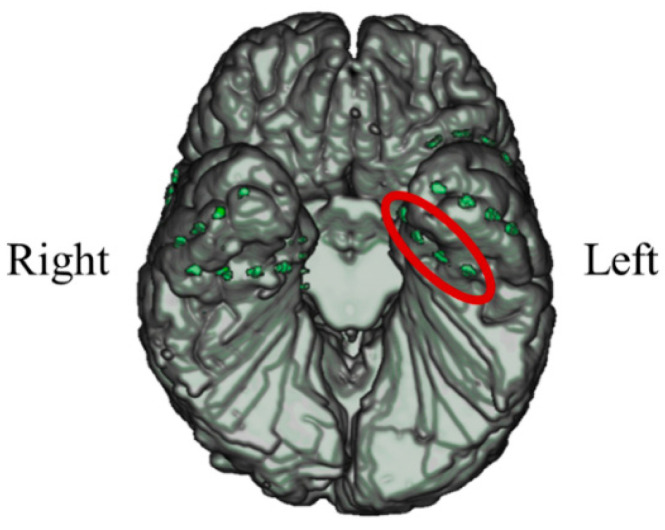
Electrodes of P03 used for neurofeedback.

**Figure 21 biomedicines-11-02262-f021:**
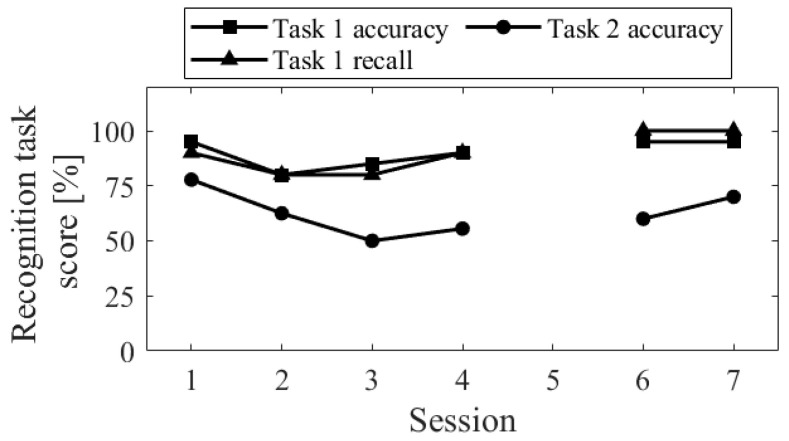
Changes in recognition task performance of P03 across sessions.

**Figure 22 biomedicines-11-02262-f022:**
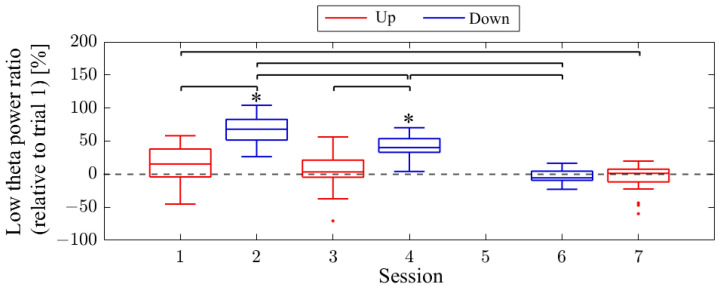
Changes in NF signals of P03 across sessions. * *p* < 0.05 compared to trial 1.

**Figure 23 biomedicines-11-02262-f023:**
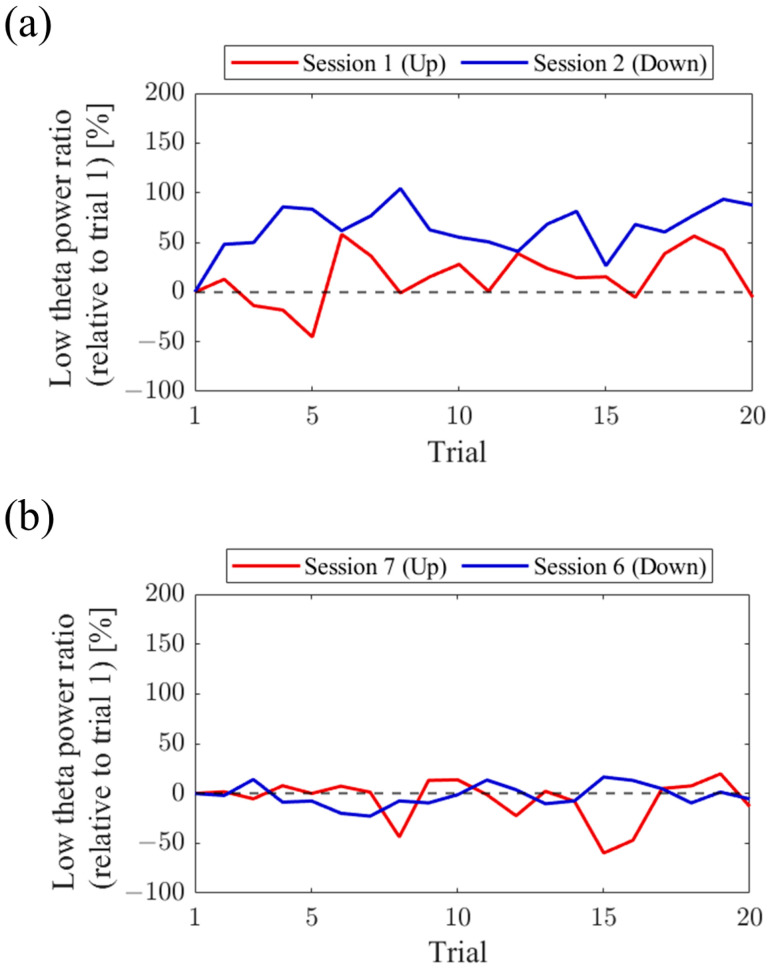
Changes of NF signal of P03 within session. (**a**) The first sessions. (**b**) The final sessions.

**Figure 24 biomedicines-11-02262-f024:**
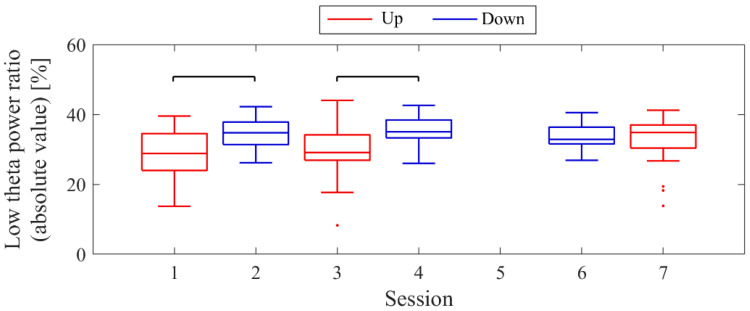
Changes of Low theta power ratio of P03 across sessions.

**Figure 25 biomedicines-11-02262-f025:**
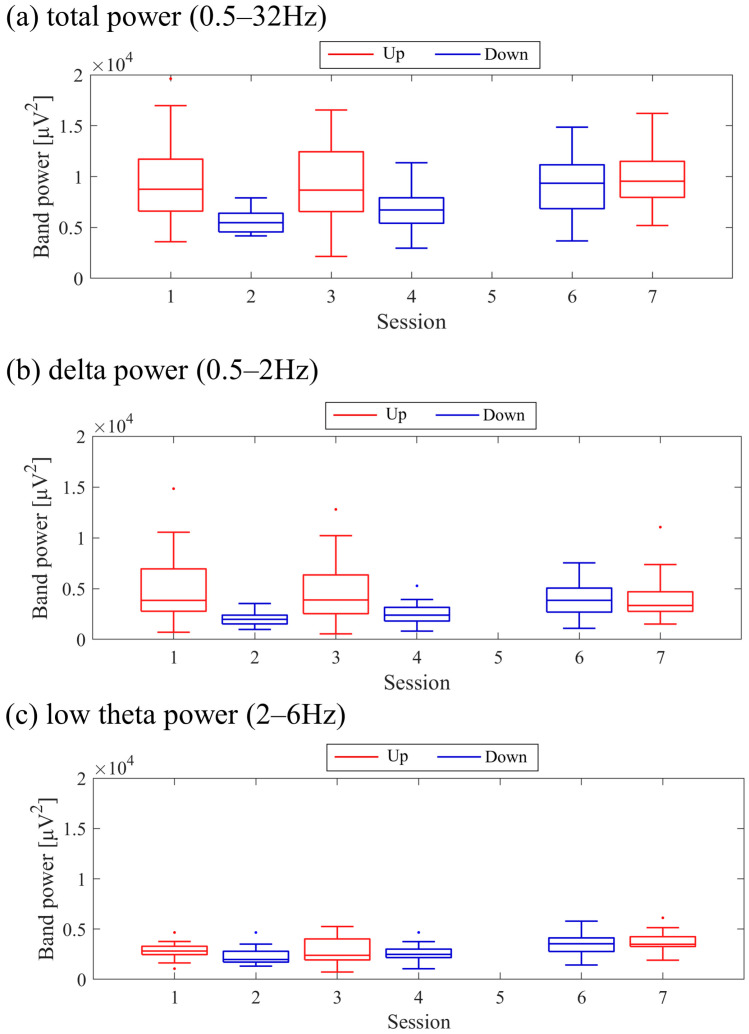
Changes in band power of P03 across sessions.

**Figure 26 biomedicines-11-02262-f026:**
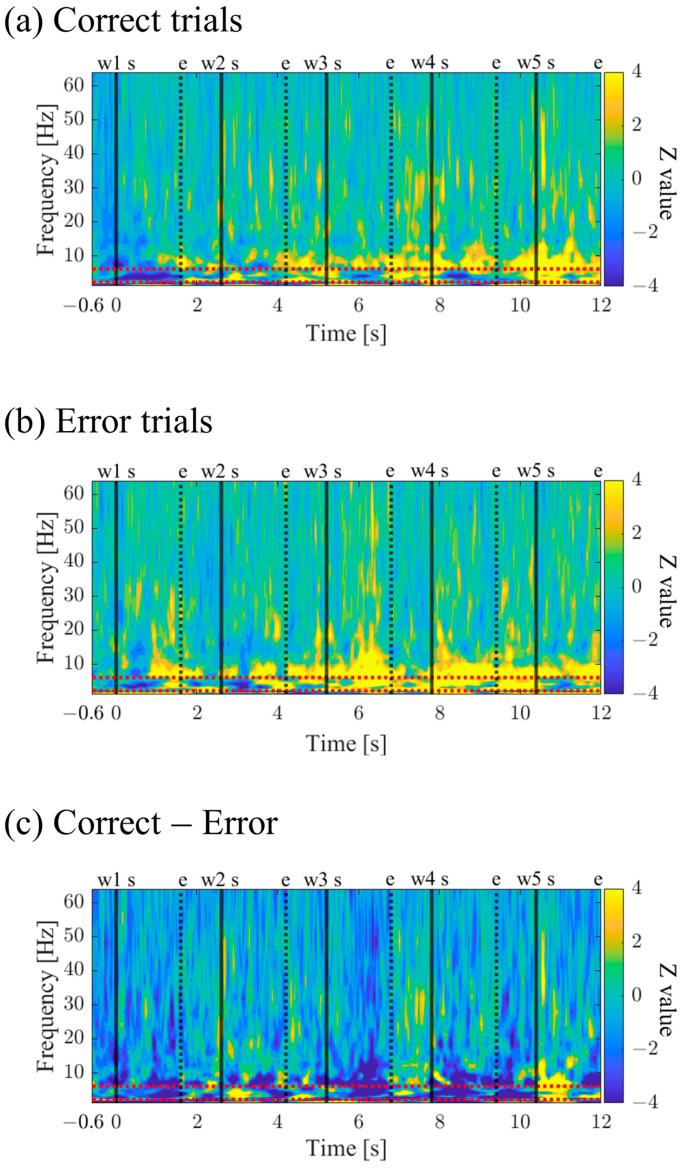
Time-frequency maps of P03 during memory encoding: (**a**) the correct trials, (**b**) the error trials, and (**c**) the difference (Correct trials–Error trials). Figure caption omitted, as it duplicates Figure 10’s caption.

**Figure 27 biomedicines-11-02262-f027:**
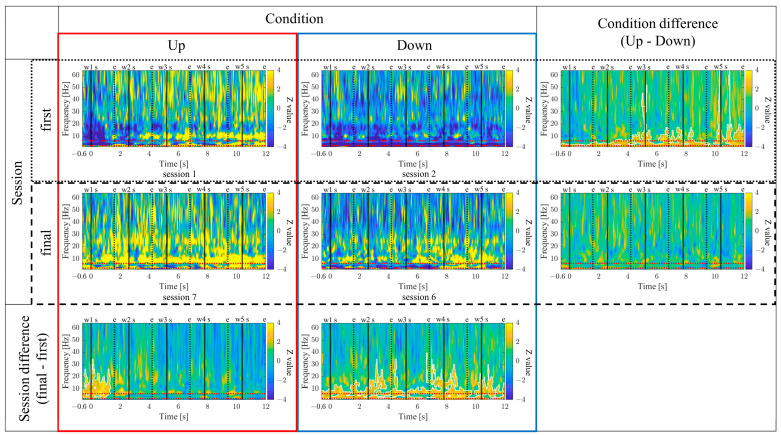
Time–frequency maps of P03 (Up/Down, the first/final sessions, and the difference). Figure caption omitted, as it duplicates Figure 10’s caption. The area enclosed by the solid white line indicates clusters of significantly larger size.

**Figure 28 biomedicines-11-02262-f028:**
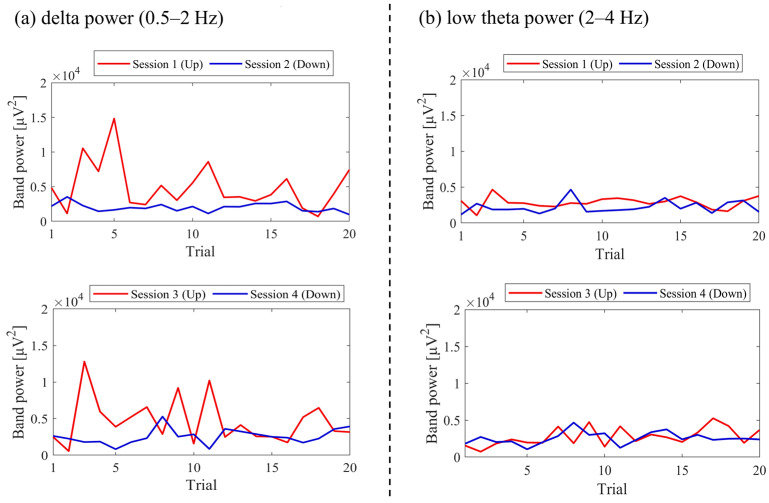
Changes in band power of P03 within session: (**a**) delta power. (**b**) low theta power.

**Table 1 biomedicines-11-02262-t001:** Participants characteristics.

ID	Age	Age at Onset	ASMs (Daily)	MRI Findings	Epileptic Focus	NF Target	FIQ	Language Laterality	WMS-R Verbal Memory Index
P01	10s	6	350 mg of PHT, 400 mg of VPA, 10 mg of CLB, 500 mg of LEV	Lt HS	Lt MTL	Rt MTL	61–70	Lt	51
P02	50s	17	275 mg of PHT, 92 mg of PB, 10 mg of CBZ, 5 mg of DZP	No lesion	Widespread	Lt MTL	71–80	Lt	102
P03	50s	27	400 mg of LCM, 200 mg of PHT, 120 mg of PB, 1000 mg of LEV, 8 mg of PER	Rt HS	Rt MTL	Lt MTL	91–100	Lt	91

ASMs, antiseizure medications; NF, neurofeedback; FIQ, full-scale intelligence quotient; WMS-R, Wechsler Memory Scale-Revised; PHT, phenytoin; VPA, valproic acid; CLB, clobazam; LEV, levetiracetam; PB, phenobarbital; CBZ, carbamazepine; DZP, diazepam; LCM, lacosamide; PER, perampanel; Lt, left; Rt, right; HS, hippocampal sclerosis; MTL, medial temporal lobe.

## Data Availability

Due to the nature of this research, participants of this study did not agree for their data to be shared publicly, so supporting data is not available.

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
