# Peer review of "Paving the Way for Memory Enhancement: Development and Examination of a Neurofeedback System Targeting the Medial Temporal Lobe"

_biomedicines, 2023, doi:10.3390/biomedicines11082262_

Round 1
Reviewer 1 Report
The paper "Paving the Way for Memory Enhancement: Development and Examination of a Neurofeedback System Targeting the Medial Temporal Lobe" investigates the influence of neurofeedback protocols in memory functions related to activity in the medial temporal lobe. To this aim the authors analyzed memory encoding performance in patients with implanted electrodes in the medial temporal lobe in conjunction with neurofeedback training. Results showed the effectiveness of neurofeedback training. The study sounds worth and timely. The discussion provides a general overview of the results but it could better insert the present study in a broader framework. For example it has ben recently shown that neurofeedback is associated with improved performance in attentional tasks (Pamplona et al 2021 NeuroImage). In addition, it has been demonstrated that neurofeedback is associated with improvement in working memory performance after the training of alpha band at temporal regions (Campos et al 2018 Front Behav Neurosci). Considering that memory and attention are tightly linked, it could be proposed that the present study identified the mnestic component of a broader cognitive processing involving also attentional load. This way it may provided a more global perspective to dynamics of which memory is part. On another account it seems that, at least in some conditions, cognitive training (not neurofeedback) is effective in improving memory tasks (Barbazzeni et al 2023 Brain Communications). It could be interesting to report these opposite results and highlight possible reasons/explanations for such differences under the light of the present study.
Author Response
We gratefully thank you for the suggestions and comments.
Please see the attachment.

Reviewer 2 Report
biomedicines (biomedicines-2493766)
Paving the Way for Memory Enhancement: Development and Examination of a Neurofeedback System Targeting the Medial Temporal Lobe
This manuscript presents a case series of 3 patients with intracranial electrodes placed into or over the mesial temporal lobe for seizure localization. The authors trained each of these patients on a task to remember words, with simultaneous neurofeedback for 2-6Hz EEG activity (referred to as “low theta”) of either UP or DOWN feedback. The authors then compared percentage band power in 2-6Hz band as compared to 0.5 to 32 Hz, as well as performance on the memory task.
The authors’ principal finding was that difference in low theta band power between the UP and DOWN conditions increases with training. No correlation with performance was identified.
This is a valuable study on a clinically important topic. Despite the limited number of patients, significant differences between the patients, and divergent results from the patients, this case series remains of interest. The authors examine the experimental details well, do not try to draw unsupported conclusions, and perhaps most importantly discuss how future studies may be improved based on their findings. The manuscript would be improved by addressing several topics as detailed below.
Other significant issues:
1. More details regarding the patient’s epilepsy/seizure history are needed. Did the occurrence of interictal activity (reduced by ASR) differ between the patients, during the trials, or was it correlated with performance? Did any seizures occur during the experiment or during the admission prior to the testing? Were the patients on antiseizure medications? Sleep-deprived?
2. Were recordings analyzed in a referential montage or something else?
3. Low theta band power is listed as 2-6Hz, isn’t this low theta and high delta?
4. Why was 0.5 to 32Hz used for total power? Intracranial recordings are typically lower in artifact and higher frequencies should be discernable.
5. Please explain the difference between Task 1 accuracy and Task 1 recall more clearly.
6. Could the cluster-based analysis used for the UP/DOWN time-frequency plots (e.g. Fig 11) be used to analyze the plots for Correct and Error plots (e.g. Fig 10)?
Minor issues:
1. Consider using the term “drug-resistant epilepsy” instead of intractable epilepsy.
2. Minor formatting issues (e.g. Page 6, Figure 1 and 2.4 Memory task line) and other figures.
3. The prose could be condensed as it is often repetitive and wordy.
3. The prose could be condensed as it is often repetitive and wordy.
Author Response

(The authors gave the same response as above.)

Round 2
Reviewer 1 Report
Accept